# AIOLI: A UNIFIED OPTIMIZATION FRAMEWORK FOR LANGUAGE MODEL DATA MIXING

**Mayee F. Chen**[1*]**, Michael Y. Hu**[2*]**, Nicholas Lourie**[3]**, Kyunghyun Cho**[2,3,4]**, Christopher Ré**[1]
[1] Computer Science Department, Stanford University; [2] Center for Data Science, NYU;
[3] Computer Science Department, NYU; [4] Prescient Design, Genentech

## ABSTRACT

Language model performance depends on identifying the optimal mixture of data groups to train on (e.g., law, code, math). Prior work has proposed a diverse set of methods to efficiently learn mixture proportions, ranging from fitting regression models over training runs to dynamically updating proportions throughout training. Surprisingly, we find that no existing method consistently outperforms a simple stratified sampling baseline in terms of average test perplexity. To understand this inconsistency, we unify existing methods into a standard framework, showing they are equivalent to solving a common optimization problem: minimize average loss subject to a method-specific *mixing law*—an implicit assumption on the relationship between loss and mixture proportions. This framework suggests that measuring the fidelity of a method's mixing law can offer insights into its performance. Empirically, we find that existing methods set their mixing law parameters inaccurately, resulting in the inconsistent mixing performance we observe. Using this insight, we derive a new online method named AIOLI, which directly estimates the mixing law parameters throughout training and uses them to dynamically adjust proportions. Empirically, AIOLI outperforms stratified sampling on 6 out of 6 datasets by an average of 0.27 test perplexity points, whereas existing methods fail to consistently beat stratified sampling, doing up to 6.9 points worse. Moreover, in a practical setting where proportions are learned on shorter runs due to computational constraints, AIOLI can dynamically adjust these proportions over the full training run, consistently improving performance over existing methods by up to 12.012 test perplexity points.

## 1 INTRODUCTION

It is important to determine what data to train on for a language model (LM) to acquire a range of capabilities, from generating code to understanding scientific literature and conversing with users (Albalak et al., 2024; Longpre et al., 2024; Li et al., 2024). To achieve this, practitioners mix data from various groups (such as code files, scientific papers, and chat logs) in specific proportions to compose an overall training dataset—a procedure known as *data mixing*. Identifying the optimal mixture proportions is critical to LLM performance. However, a brute-force trial-and-error search over the proportions is computationally expensive, requiring many training runs.

Recent work introduces two types of data mixing algorithms that *learn* mixture proportions: offline and online methods. Offline methods conduct multiple training runs with varying proportions, fit a regression model to predict performance, and use this model to determine the optimal static mixture (Ye et al., 2024; Liu et al., 2024). Online methods adjust the mixture proportions dynamically throughout training using information from the model, such as its loss and gradients (Chen et al., 2023; Fan et al., 2024; Xie et al., 2023a; Albalak et al., 2023). All mixing methods require at least one training run to learn the proportions but are more efficient than a brute-force search.

Given the wide range of methods available, it is important to determine which ones are effective. However, when we evaluated existing methods, we found that *no method consistently outperformed stratified sampling*—a simple baseline that uniformly mixes groups and requires zero extra training runs—across all sets of data groups in terms of average test perplexity (Table 2). This surprising outcome suggests that all existing methods suffer from some common weaknesses. To make progress in data mixing, we identify three objectives: 1) improve our **understanding** of the underlying assumptions of existing methods, 2) assess the **fidelity** of these assumptions in practice to better understand performance, and 3) apply our insights to develop principled **new data mixing methods**.

---

[*]Equal contribution. Contact: mfchen@stanford.edu, michael.hu@nyu.edu

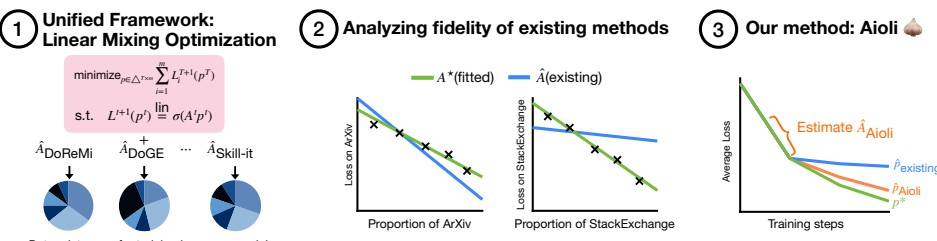

Figure 1: Left: existing methods can be expressed in a unified optimization framework, in which they implicitly assume a linear or log-linear loss-proportion relationship. Center: the (log)-linear parameterizations are well-specified, but existing methods set their parameters incorrectly. Right: AIOLI, an online mixing method that more accurately estimates the parameters that capture the true loss-proportion relationship.

In this paper, we improve our **understanding** of data mixing methods by showing that many existing methods can be expressed in a unified optimization framework, which we call Linear Mixing Optimization (LMO) (Section 3). These methods are equivalent to solving an optimization problem that sets proportions to minimize the average loss per data group, subject to an implicit method-dependent *mixing law*—an assumption relating loss per group and mixture proportions. We find that all current mixing laws share the same parameterization: for training round $t$ from 1 to $T$,

$$L^{t+1}(p^t) \stackrel{\text{lin}}{=} \sigma(A^t p^t),$$

where $p^t \in \triangle^m$ (the simplex) are mixing proportions over $m$ given data groups at time $t$, $L^{t+1}(p^t) : \triangle^m \to (\mathbb{R}^+)^m$ are the losses per group at the next timestep, $A^t \in \mathbb{R}^{m \times m}$ is a parameter matrix, $\sigma = \text{Id}$ or $\exp$, and $\stackrel{\text{lin}}{=}$ means equal up to linear transformation. Existing offline methods assume a static ($T = 1$) log-linear parameterization of the mixing law, while online methods assume a linear dynamic mixing law. All methods set the parameters of their mixing laws differently (Table 1), and offline methods solve the optimization problem directly while online methods solve it greedily using exponentiated gradient descent. Our framework reveals the underlying assumptions of each method in terms of the mixing law's parameterization, the values of the parameters, and how the optimization problem is solved. Furthermore, the fidelity of the mixing law and solving strategy dictates the optimality of the method, providing us with a new tool for understanding data mixing methods.

Applying the LMO framework, we test the **fidelity** of existing methods' assumptions, examining if they hold in practice (Section 4). Both the log-linear static and linear dynamic parameterizations capture the true loss-proportion relationship across datasets, achieving an average of 0.0005 MSE and 0.969 $R^2$. We then show that although existing mixing laws are well-specified, methods can set their parameters ($A^t$) inaccurately, causing poor performance. We compare each method's parameters to the optimal parameters, which we approximate by fitting the mixing laws to training runs. We find that the method's parameters can differ significantly from the optimal parameters, and the extent of these deviations is correlated with method performance relative to stratified sampling (Figure 3), helping explain our initial observations. Finally, we validate the assumptions used in solving the optimization problem, finding that the greedy approximation in online methods is a reasonable proxy for the full objective. Our analysis shows that existing methods' parameterizations and solving strategies are of high fidelity, but their parameters are not.

To validate these insights, we develop AIOLI, a **simple new online data mixing method** derived from the LMO framework (Section 5). Unlike existing online methods, AIOLI directly estimates the parameters $A^t$ from the current training run by fitting the mixing law on the history of losses and dynamic mixture proportions so far. AIOLI is thus able to dynamically adjust proportions without requiring any extra training runs.

We evaluate AIOLI in two settings by training 160M models on various combinations of data sources from SlimPajama (Soboleva et al., 2023) (Section 6). First, we compare AIOLI to existing data mixing methods and find that AIOLI consistently outperforms stratified sampling on all 6 datasets, by an average of 0.274 and up to 0.439 points in test perplexity. On the other hand, existing data mixing methods do worse than stratified on at least one dataset by up to 6.9 perplexity points, despite using extra training runs. As we expect, the parameters of AIOLI are also more consistently close to the optimal parameters (Figure 2). Second, we consider a scenario with limited additional computational resources, in which practitioners cannot run experiments for learning mixture proportions for the full training duration. In this setting, mixture proportions learned on a shorter run may not perform well on the longer final run. We find that using AIOLI to dynam-

ically adjust these learned proportions throughout the final training run can improve performance by an average of 1.202 perplexity points in 28 out of 30 cases, compared to using the learned proportions directly.

## 2 PROBLEM SETUP

We formalize the data mixing problem and establish notation. In data mixing, we have $m$ data groups of text, such as GitHub, BooksCorpus, and arXiv. We are given train, validation, and test sets for each data group, which we denote as $D_{\text{train}}^i, D_{\text{val}}^i, D_{\text{test}}^i$ for the $i$th group. Define $D_{\text{train}} = \{D_{\text{train}}^1, ..., D_{\text{train}}^m\}$, and similarly define $D_{\text{val}}$ and $D_{\text{test}}$.

**Data & Mixing.** During training, we show the model a total of $N$ examples from $D_{\text{train}}$ over $S$ training steps. To express how data proportions can change throughout training, we divide training into $T$ equal rounds. Each round $t$ uses a mixture proportion from the probability simplex: $p^t = [p_1^t, ..., p_m^t] \in \triangle^m$. *Static mixtures* use only a single round ($T = 1$): $\boldsymbol{p} = (p^1)$, while *dynamic mixtures* use several ($T > 1$): $\boldsymbol{p} = (p^1, ..., p^T)$.

**Model & Loss.** Let $f(\boldsymbol{p}, t)$ refer to the language model, $f$, at the beginning of round $t$ where the model has been trained on data sampled using mixture proportions $p^1, \cdots, p^{t-1}$ so far. Given a model $f$, we can compute its loss on each group using the training data, $L_{\text{train}}(f) = (L_{\text{train},1}(f), ..., L_{\text{train},m}(f))$, and similarly with the validation, $L_{\text{val}}(f)$, and test data, $L_{\text{test}}(f)$. In this notation, the loss at the end of training can be expressed as $L_{(\cdot)}(f(\boldsymbol{p}, T+1))$. When the $f$ being referred to is obvious, we simply write $L_{(\cdot)}^t(\boldsymbol{p})$, and for static mixtures we drop the superscript: $L_{(\cdot)}(\boldsymbol{p})$.

**Data Mixing Problem.** Given a set of data groups, an LM $f$ to train for $S$ steps with $N$ samples, and $T$ rounds of training (i.e., whether we use static or dynamic proportions), we aim to determine the $\boldsymbol{p}$ that minimizes the total test loss across groups: $\text{minimize}_{\boldsymbol{p} \in \triangle^{T \times m}} \sum_{i=1}^m L_{\text{test},i}^{T+1}(\boldsymbol{p})$.

This objective aims to produce a trained model that does well on many data groups, which can serve as a proxy for downstream performance. However, without assuming additional structure on $L^{T+1}(\boldsymbol{p})$, this problem can only be solved with a brute-force search over $\boldsymbol{p}$, which requires training many different models. In the next section, our LMO framework imposes a constraint on $L^{t+}(\boldsymbol{p})$ that allows many existing methods to be expressed as approaches to solving this problem.

## 3 A UNIFIED OPTIMIZATION FRAMEWORK FOR DATA MIXING

We introduce the LMO framework by stating the general optimization problem (Section 3.1). Then, we show how this framework can express several existing methods (Section 3.2, 3.3), with a summary of our insights regarding these methods in Section 3.3.3.

### 3.1 LINEAR MIXING OPTIMIZATION (LMO) FRAMEWORK

The LMO framework consists of an optimization problem that is equivalent to the data mixing minimization problem (Section 2), subject to an additional constraint:

$$\text{minimize}_{\boldsymbol{p} \in \triangle^{T \times m}} \sum_{i=1}^m L_{\text{val},i}^{T+1}(\boldsymbol{p}) \tag{1}$$

$$\text{s.t.} \ L_{\text{val},i}^{t+1}(\boldsymbol{p}) = c_i^t + b_i^t \sigma\Big(\sum_{j=1}^m -A_{ij}^t p_j^t\Big) \ \forall i \in [m], t \in [T] \tag{2}$$

for some $A^t, b^t, c^t$, and $\sigma$. $A^t \in \mathbb{R}^{m \times m}$ is a matrix that encodes cross-group interactions, where $A_{ij}^t$ intuitively describes how much training on group $j$ at $t$ impacts group $i$'s loss. $b^t, c^t \in \mathbb{R}^m$ are group-specific parameters. $\sigma : \mathbb{R} \to \mathbb{R}$ is either the identity function (Id) or the exponential function (exp). We refer to the constraint in (2) as a *mixing law* that specifies the assumed relationship between loss and proportions.

There are three components of this problem that need to be specified to yield a way to set $\boldsymbol{p}$: a) the parameterization of the mixing law ($T$, $\sigma$), b) the values of the parameters ($A^t, b^t, c^t$), and c) how to solve the problem. We express existing methods in LMO by specifying these components.

| Method | 1) Mixing Law Parameterization | 2) Parameters | 3) Solver |
|---|---|---|---|
| DML | $L_{\text{val},i}(\boldsymbol{p}) = c_i + b_i \exp\left(\sum_{j=1}^{m} -A_{ij}p_j\right)$ | Fit from $\geq m+1$ training runs | Direct |
| Skill-It | $L_{\text{val},i}^{t+1}(\boldsymbol{p}) = L_{\text{val},i}^{t}(\boldsymbol{p}) - b^t \sum_{j=1}^{m} A_{ij}^t p_j^t$ | $A_{ij}^t = L_{\text{val},i}^t(\boldsymbol{p})(L_{\text{val},i}^{T+1}(\mathbf{1}_j) - L_{\text{val},i}^{1}(\mathbf{1}_j))/L_{\text{val},i}^{1}(\mathbf{1}_j)$ | EGD |
| DoReMi | $L_{\text{val},i}^{t+1}(\boldsymbol{p}) = L_{\text{val},i}^{t}(\boldsymbol{p}) - b^t \sum_{j=1}^{m} A_{ij}^t p_j^t$ | $A_{ii}^t = \min\{L_{\text{train},i}^t(\boldsymbol{p}) - L_{\text{train},i}(f_{\text{ref}}), 0\}$ | EGD |
| DoGE | $L_{\text{val},i}^{t+1}(\boldsymbol{p}) = L_{\text{val},i}^{t}(\boldsymbol{p}) - b^t \sum_{j=1}^{m} A_{ij}^t p_j^t$ | $A_{ij}^t = \langle \nabla L_{\text{val},i}^t(\boldsymbol{p}), \nabla L_{\text{train},j}^t(\boldsymbol{p}) \rangle$ | EGD |
| AIOLI | $L_{\text{val},i}^{t+1}(\boldsymbol{p}) = L_{\text{val},i}^{t}(\boldsymbol{p}) - \sum_{j=1}^{m} A_{ij}^t p_j^t$ | Fit from history of $L_{\text{val}}$ and $\boldsymbol{p}$ | EGD |

Table 1: Summary of how existing methods and AIOLI are expressed in the LMO framework (1).

## 3.2 PRELIMINARIES FOR UNIFYING METHODS

We discuss preliminaries before presenting existing methods and explaining how they can be expressed in the LMO framework. First, we formally define what it means for a method to be *expressed* in the LMO framework. Then, we present a result that allows us to convert between linear dynamic mixing laws and a way to set $\boldsymbol{p}$, which we will to use to express online methods in our framework in Section 3.3.

**Definition 1.** *We say that a data mixing method can be **expressed** in the LMO framework if its exact algorithm—how it sets proportions $\boldsymbol{p}$ and trains model $f$ in terms of $\boldsymbol{p}$—can be equivalently constructed by specifying a mixing law and way of solving the LMO optimization problem.*

This definition allows us to cast existing methods as a way of solving the LMO optimization problem based on how they set $\boldsymbol{p}$ and train according to $\boldsymbol{p}$, even if the methods themselves are not originally designed to minimize average test loss.

**Converting mixing laws into update rules.** When $T > 1$, a natural way to solve the LMO optimization problem is via exponentiated gradient descent (EGD) (Kivinen & Warmuth, 1997; Arora et al., 2012), which updates $p^t$ greedily while ensuring that it remains on the probability simplex. The following lemma presents the EGD update rule for the LMO optimization problem when $\sigma = \text{Id}$.

**Lemma 1.** *The EGD update rule for (1) subject to $L_{val,i}^{t+1}(\boldsymbol{p}) = c_i^t - b_i^t \sum_{j=1}^{m} A_{ij}^t p_j^t \ \forall i \in [m]$ is*

$$p_j^{t+1} = \frac{1}{Z^t} \cdot p_j^t \exp\left(\eta \sum_{i=1}^{m} b_i^t A_{ij}^t\right) \forall j \in [m], \tag{3}$$

*where $\eta > 0$ is the step size and $Z^t$ is a normalizing constant such that $p_j^{t+1} \in \triangle^m$.*

This lemma shows how to adjust $p^t$ dynamically to solve the LMO optimization problem. Notably, this update rule is defined in terms of the mixing law parameters, $A^t$ and $b^t$. This gives us a way to convert between how a method sets $\boldsymbol{p}$ and the implicit *assumption* it makes in the mixing law.

## 3.3 UNIFYING EXISTING METHODS

We discuss four existing data mixing methods and express them as specific instances of the LMO framework. A summary of our insights is provided in Section 3.3.3 and Table 1. In Appendix B.1, we comment on how several other online and offline data mixing methods are related to our framework, and all proofs for this section are in Appendix B.2.

### 3.3.1 OFFLINE METHODS

**Data Mixing Laws (DML).** Ye et al. (2024) propose an offline method using a static mixing law ($T = 1$): $L_{\text{val},i}(\boldsymbol{p}) = c_i + b_i \exp(\sum_{j=1}^{m} -A_{ij}p_j)$ for $i \in [m]$, with $A, b, c$ learned by sweeping training runs over static proportions ($\geq m+1$ runs to avoid being underdetermined). They select the proportion that minimizes the predicted validation loss. This law can be derived from (2) with $\sigma = \exp$, showing that LMO with a) log-linear static mixing law, b) fitted parameters, and c) direct computation of $\boldsymbol{p}$ can express DML.

### 3.3.2 ONLINE METHODS

We provide a colloquial description and an algorithmic description of the following three online methods. Then, in Theorem 1 we demonstrate how they all are expressed in LMO using a linear dynamic mixing law, the EGD update rule, and method-specific mixing law parameters.

**Skill-It.** Chen et al. (2023) is an online method motivated by curriculum learning that dynamically adjusts mixture proportions. Data group interactions are expressed in a "skills graph," where each edge denotes how much the loss on one group changes when trained on another. The skills graph is learned in advance using $m$ training runs and then used to update proportions $p^t$ throughout training.

Concretely, the skills graph matrix $A^{\text{SG}}$ has entries $A_{ij}^{\text{SG}} = (L_{\text{val},i}^{T+1}(\mathbf{1}_j) - L_{\text{val},i}^1(\mathbf{1}_j))/L_{\text{val},i}^1(\mathbf{1}_j)$ indicating the relative decrease in loss on group $i$ when training a model on group $j$ only. This is used in the Skill-It update rule, $p_j^{t+1} \propto p_j^t \exp(\eta \sum_{i=1}^m A_{ij}^{\text{SG}} L_{\text{val},i}^t(\boldsymbol{p}))$ for all $j \in [m]$ and learning rate $\eta > 0$. This rule determines $p^{t+1}$, which is then used to sample $D_{\text{train}}$ for training $f$ in the next round.

**DoReMi.** Xie et al. (2023a) is an online method that applies ideas from distributionally robust optimization to data mixing, where the training objective minimizes the worst-group excess loss over a model trained with stratified sampling. $p^t$ is updated dynamically to minimize this excess loss and then averaged for the final run. DoReMi requires two additional runs to learn a static $\boldsymbol{p}$.

Concretely, let $f_{\text{ref}} = f(\text{Unif}(m), T + 1)$ denote a "reference model" that is first trained using stratified sampling. Then, a "proxy model" uses dynamic proportions according to the update rule $p_j^{t+1} \propto p_j^t \exp(\eta \max\{L_{\text{train},j}^t(\boldsymbol{p}) - L_{\text{train},j}(f_{\text{ref}}), 0\})$ for all $j \in [m]$ and step size $\eta > 0$. This $p^{t+1}$ is used to weight the training objective, such that the proxy model is updated to minimize $\sum_{i=1}^m p_i^{t+1} L_{\text{train},i}(f)$ at the next timestep. The averaged static proportions $\frac{1}{T} \sum_{t=1}^T p^t$ are then used in the final run.

**DoGE.** Fan et al. (2024) is an online method that solves a bi-level optimization problem in which $p^t$ is updated to minimize the average training loss at each step. By using a first-order Taylor approximation of the training loss, $p^t$ is updated using the gradient of each data group. The dynamic proportions are then averaged for the final run. DoGE requires one additional run to learn a static $\boldsymbol{p}$.

Concretely, a proxy model is trained using $p_j^{t+1} \propto p_j^t \exp(\eta \langle \nabla L_{\text{train},j}(f^t), \sum_{i=1}^m \nabla L_{\text{val},i}(f^t) \rangle)$, and $f$ is updated to minimize the training loss weighted by $p^t$, similar to DoReMi. The averaged static proportions $\frac{1}{T} \sum_{t=1}^T p^t$ are used in the final run.

**Framework expression.** All three online methods use an update rule $p_j^{t+1} \propto p_j^t \exp(\cdot)$, which is similar to (3). This provides intuition for our main theorem, which expresses these methods in LMO.

**Theorem 1.** *Define the following parameters for each method:*

- $A^{t,\textit{Skill-It}} \in \mathbb{R}^{m \times m}$, *where* $A_{ij}^{t,\textit{Skill-It}} = L_{val,i}^t(\boldsymbol{p})(L_{val,i}^{T+1}(\mathbf{1}_j) - L_{val,i}^1(\mathbf{1}_j))/L_{val,i}^1(\mathbf{1}_j)$ *for all* $i,j \in [m]$,

- $A^{t,\textit{DRM}} \in \mathbb{R}^{m \times m}$, *where* $A_{ii}^{t,\textit{DRM}} = \min\{L_{train,i}^t(\boldsymbol{p}) - L_{train,i}(f_{ref}), 0\}$ *and* $A_{ij}^{t,\textit{DRM}} = 0$ *for* $i \neq j$,

- $A^{t,\textit{DoGE}} \in \mathbb{R}^{m \times m}$, *where* $A_{ij}^{t,\textit{DoGE}} = \langle \nabla L_{val,i}^t(\boldsymbol{p}), \nabla L_{train,j}^t(\boldsymbol{p}) \rangle$ *for all* $i,j \in [m]$.

*Instantiating the LMO framework (1) with a) a linear dynamic mixing law* $L_{val,i}^{t+1}(\boldsymbol{p}) = L_{val,i}^t(\boldsymbol{p}) - b^t \sum_{j=1}^m A_{ij}^t p_j^t$, *b) parameters* $A^t = A^{t,\textit{Skill-It/DRM/DoGE}}$, *and c) EGD to solve for* $\boldsymbol{p}$ *allows for us to express Skill-It, DoReMi, and DoGE, respectively.*

### 3.3.3 SUMMARY OF LMO FRAMEWORK INSIGHTS

Table 1 summarizes how existing methods are expressed in the LMO framework. LMO reveals the assumptions each method makes through how the components of the framework are specified. First, all mixing laws are either linear or log-linear. Second, the mixing laws differ in the values of the parameters used. For example, Skill-It's $A^t$ is the current loss times a static skills graph matrix, while DoReMi's $A^t$ is diagonal. Third, offline mixing methods solve for $\boldsymbol{p}$ directly while online mixing methods use EGD, which uses a greedy approximation. If the mixing law and solving strategy assumptions hold true in practice, then the method yields optimal mixture proportions. In the next section, we study the fidelity of these assumptions.

## 4 ANALYZING FIDELITY OF EXISTING METHODS WITH THE LMO FRAMEWORK

We examine the fidelity of the assumptions made by existing methods in terms of the three components of the LMO framework: a) the mixing law parameterization, b) values of the mixing law parameters, and c) how to solve the optimization problem for $\boldsymbol{p}$. After providing experiment details (Section 4.1), we discuss these three components in order (Section 4.2-4.4).

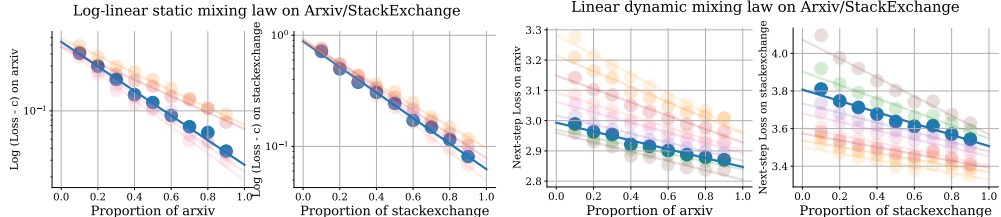

Figure 2: Left: $p_i$ vs $\log(L_{\text{val},i}(\boldsymbol{p})-c_i)$ with fitted static log-linear mixing law. Right: $p_i^t$ vs $L_{\text{val},i}(\boldsymbol{p})$ with fitted linear dynamic mixing law. Colors represent random seeds (left) and initial $p^0 \in \mathcal{P}$ (right, blue is 0.7,0.3). Both laws fit the true loss-proportion relationship well.

## 4.1 Experiment Details

**Data settings.** We use a sampled version of SlimPajama (Soboleva et al., 2023; Yoon, 2023), a pre-processed version of the RedPajama pretraining dataset (Together.ai, 2023). SlimPajama consists of 7 data groups: ArXiv, Books, CommonCrawl, C4 (Raffel et al., 2019), Github, StackExchange, and Wikipedia. To develop a fine-grained understanding of data mixing, we create 6 settings by extracting combinations of these groups. We study three settings with $m = 2$: Arxiv/Stackexchange, Github/C4, and Book/StackExchange. We study two settings with $m = 3$: Arxiv/Book/StackExchange and CommonCrawl/Github/Wikipedia. Finally, we study mixing over the full SlimPajama dataset with $m = 7$.

**Models.** We train 160M parameter GPT-style decoder-only LLMs with batch size 8 and context length 2048. For $m = 2, 3$, we train for 5K steps, and for $m = 7$, we train for 40K steps.

**Training sweeps.** To assess the true loss-proportion relationship and compare it to the assumptions made by existing methods, we conduct training sweeps over different mixture proportions, denoted as $\mathcal{P}$. For $m = 2$, we set $\mathcal{P} = \{[0.1,0.9],[0.2,0.8],...,[0.9,0.1]\}$. For $m = 3$ and 7, we set $\mathcal{P}$ equal to 10 $\boldsymbol{p}$'s and 40 $\boldsymbol{p}$'s drawn from the Dirichlet distribution with $\alpha = 1.0$ and 1.5, respectively.

## 4.2 Mixing law parameterization

We examine whether existing methods' mixing law parameterizations—log-linear static and linear dynamic—capture the true loss-proportion relationship. By empirically fitting them to loss-proportion pairs, we find that both parameterizations are indeed well-specified. Full results for both mixing laws are in Table 5 in Appendix C.1. We discuss the generality of these parameterizations across training scales and other datasets, as well as higher-order parameterizations, in Appendix C.1.1.

**Setup.** For the *log-linear static mixing law*, we study if there exists $A, b, c$ such that $L_{\text{val},i}(\boldsymbol{p})$ can be expressed as $c_i + b_i \exp(\sum_{j=1}^{m} -A_{ij}p_j)$ for all $i \in [m]$. We fit the parameters using full training runs on $\mathcal{P}$. For the *linear dynamic mixing law*, we study if there exists $A^t$ such that $L_{\text{val},i}^{t+1}(\boldsymbol{p})$ can be expressed as $L_{\text{val},i}^t(\boldsymbol{p}) - \sum_{j=1}^{m} A_{ij}^t p_j^t$, for all $i \in [m]$ ($b^t$ is absorbed into $A^t$). To fit $A^t$, we select a timestep $t$ and train on a static proportion $p^0 \in \mathcal{P}$ for all $p^1,...,p^t$ until time $t$, and at $t+1$ we sweep the values of $p^{t+1} \in \mathcal{P}$.

**Results.** On average across our 6 data settings, the mean squared error (MSE) of the fitted log-linear static mixing law is $8.9 \times 10^{-4}$, and the $R^2$ coefficient of determination is 0.991. The average MSE of the fitted linear dynamic mixing law is $1.0 \times 10^{-4}$ and the $R^2$ is 0.947. See Figure 2 for examples. Since both parameterizations have high $R^2$ and low MSE, we conclude that they capture the true loss-proportion relationship well and are of high fidelity.

## 4.3 Values of mixing law parameters

As shown in Table 1, each method sets the parameters of its mixing law differently. We study how close the method-specific parameters are to the optimal parameters that are obtained when fitting the method's mixing law to the true loss-proportion relationship, and if these parameter disparities are reflected in method performance. We find that existing methods' differences in mixing law parameters are largely responsible for their performance. We omit studying DML since its parameters are fitted from full training runs and hence differ from the optimal in estimation error only.

**Setup.** For Skill-It, DoReMi, and DoGE, we select a step $t$ and obtain the method-specific $A^t$. We then sweep $\mathcal{P}$ for the next round $t+1$. This sweep is used to approximate an optimal $A^{t\star}$ that captures the true loss-mixture relationship, $L^{t+1}_{\text{val}}(\boldsymbol{p}) = L^t_{\text{val}}(\boldsymbol{p}) - A^{t\star}p^t$, as well as fit a $b^t \in \mathbb{R}$ used for scaling $A^t$ (details in Appendix C.2). We study the relationship between $\tilde{A}^t := b^t A^t$ and $A^{t\star}$, and how it is related to the performance of the method.

To express similarity between $\tilde{A}^t$ and $A^{t\star}$ in a way that is reflected in performance, we observe that from Lemma 1, $p^t$ is updated using the column sum of $A^t$, $\mathbf{1}^\top A^t$. Moreover, the magnitude of $A^t$ is not critical to performance since the step size $\eta$ can always be tuned to control this. Therefore, we compare the vectors $\tilde{a}^t = \mathbf{1}^\top \tilde{A}^t / \|\mathbf{1}^\top \tilde{A}^t\|_2$ and $a^{t\star} = \mathbf{1}^\top A^{t\star} / \|\mathbf{1}^\top A^{t\star}\|_2$. Finally, we note that the order of the elements of $\tilde{a}^t$ determines the update direction from $p^t$ to $p^{t+1}$ in Lemma 1. Therefore, we propose a similarity score that is an average of cosine similarity and the Spearman rank correlation, $\text{sim}(\tilde{A}^t, A^{t\star}) = 0.5\text{cossim}(\tilde{a}^t, a^{t\star}) + 0.5\text{Spearman}(\tilde{a}^t, a^{t\star})$. This metric is bounded between $-1$ and $1$, where $1$ indicates $\tilde{a}^t = a^{t\star}$ and $-1$ indicates $\tilde{a}^t = -a^{t\star}$.

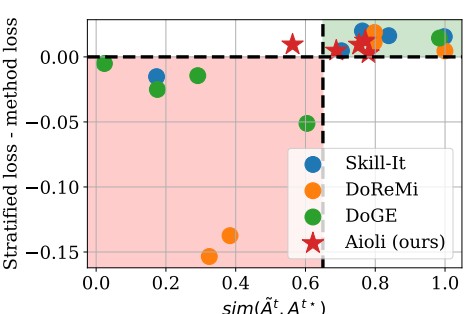

Figure 3: Improvement over stratified sampling versus optimality of $A^t$. Each dot represents a method applied to a dataset. The red region shows that existing methods are worse than stratified on at least 1 dataset. The vertical dashed line serves as a visual aid.

**Results.** In Figure 3, we plot each method's $\text{sim}(\tilde{A}^t, A^{t\star})$ versus each method's improvement over the stratified sampling baseline, which sets $p_i = 1/m$ for all $i \in [m]$, for each dataset in the $m = 2,3$ data settings. We find that no existing online method works well across all datasets (also see Table 2), and that our metric and loss improvement have a moderate positive correlation ($R^2 = 0.491$). This suggests that $A^t$'s accuracy is critical to the performance of online methods, and that existing methods' $A^t$ are not consistently accurate across the datasets. In Appendix C.2.1, we give more details on the structure of $A^{t\star}$, providing intuition for why existing methods' parameters cannot express it.

## 4.4 SOLVING STRATEGY

We study the assumptions made in how existing methods solve the LMO optimization problem. We find that the greedy approximation used by EGD, $\text{minimize}_{p^t} \sum_{i=1}^m L^{t+1}_{\text{val},i}(\boldsymbol{p})$, does not significantly compromise performance compared to full optimization of dynamic proportions, which has an exponentially large solution space. In particular, we study if greedily selecting $p^t$ from $\mathcal{P}$ at each $t$ yields the optimal dynamic proportions in $\mathcal{P}^T$, and we find that this holds in 2 out of 3 data settings (Table 8). This suggests that the greedy approximation can simplify optimization without substantial performance loss. We also comment on other possible solving strategies in Appendix C.3.

## 5 AIOLI: A METHOD FOR IMPROVED DATA MIXING

To validate our insights from Section 4, we develop AIOLI, an online method derived from the LMO framework. We have three takeaways from section 4:

a) A linear dynamic mixing law, $L^{t+1}_{\text{val},i}(\boldsymbol{p}) = L^t_{\text{val},i}(\boldsymbol{p}) - \sum_{j=1}^m A^t_{ij}p^t_j$ for all $i \in [m]$, can capture the loss-proportion relationship with high fidelity (Section 4.2).

b) Existing online methods often set the parameters $A^t$ to be very different from true $A^{t\star}$ (Section 4.3).

c) Exponentiated gradient descent can recover near-optimal performance while simplifying the optimization problem, avoiding an exponential solution space (Section 4.4).

We thus directly specify the linear dynamic mixing law parameterization and EGD as two out of three LMO components of AIOLI since we found that their assumptions generally hold in practice. According to Lemma 1, the update rule given these two components is $p^{t+1}_j \propto p^t_j \exp(\eta \sum_{i=1}^m A^t_{ij})$ ($b^t$ is absorbed into $A^t$). Thus, our primary mandate in creating AIOLI is to construct and utilize an $A^t$ that is an accurate estimate of the true $A^{t\star}$ in the linear dynamic mixing law, which existing online methods fail to achieve.

**Estimating $A^{t\star}$.** To build intuition, we first consider a high-cost naive approach. For each round, we could conduct a training sweep of $m$ different proportions $p^{t,1},...,p^{t,m}$, and observe each resulting change in loss. We could then solve a system of $m$ equations for each $i$: $L_{\text{val},i}^t - L_{\text{val},i}^{t+1}(p^{t,s}) = \sum_{j=1}^m A_{ij}^t p_j^{t,s}$ for $s \in [m]$, obtaining vectors $A_1^t,...,A_m^t$. However, this approach effectively requires $m$ extra training runs. AIOLI similarly solves a system of equations, but it computes loss changes per sweep mixture without requiring extra training. First, it allocates $\delta$ fraction of the training round for learning $A^t$. Second, it partitions this $\delta$ into $K = mk$ intervals and trains according to an interleaved order on $p^{t,1},...,p^{t,m}$. After training on each $p^{t,j}$, we record the resulting change in validation losses, and we average over all of $p^{t,j}$'s intervals. Intuitively, the interleaving ensures that the model is trained on each $p^{t,j}$ for several intervals throughout $\delta$, which can approximate if we were to train on $p^{t,j}$ for the entire $\delta$ (which approximates the entire round). This procedure is outlined in LEARNPARAMS (Alg. 2), with more details in Appendix D and Figure 7.

**AIOLI.** First, we set $p^0$ to be uniform. In each round, we estimate $A^t$ using LEARNPARAMS and then normalize the entries of $A^t$, producing $\bar{A}^t$. Otherwise, $A^t$ decreases along with loss over time, resulting in the first few $p^t$ updates being much larger in magnitude than others. Then, we update the proportions using $p_j^t \propto p_j^{t-1} \exp(\eta \sum_{i=1}^m \bar{A}_{ij}^t)$, as in Lemma 1, and train for the remainder of that round using $p_j^t$.

Finally, we design AIOLI so that it can also be used to improve other data mixing methods, which we study in Section 6.2. Mixture proportions can be updated using AIOLI either from the start of training or from the middle of a run. In the latter case, we denote an initial static mixture $\boldsymbol{p}^{\text{init}} \in \triangle^m$ and initial number of steps $S_{\text{init}}$. If $S_{\text{init}}$ is nonzero, AIOLI trains according to $\boldsymbol{p}^{\text{init}}$ for the first $S_{\text{init}}$ steps before beginning to update the mixture proportions. AIOLI is presented in Algorithm 1.

---

**Algorithm 1** AIOLI

1: **Input:** data $D_{\text{train}}$, $D_{\text{val}}$, model $f^1$. Initial steps $S_{\text{init}}$, initial proportions $\boldsymbol{p}^{\text{init}} \in \triangle^m$. $T$ rounds over $S - S_{\text{init}}$ remaining steps, $\delta$ fraction per round for learning parameters, learning rate $\eta$, one-hot smoothing factor $\varepsilon$.
2: If $S_{\text{init}} \neq 0$, train $f^1$ on $\boldsymbol{p}^{\text{init}}$ for $S_{\text{init}}$ steps.
3: Set $p^0 = \text{Unif}(m)$.
4: **for** $t = 1,...,T$ **do**
5:     Set $A^t, f^{t+\delta} \leftarrow$ LEARNPARAMS $(D_{\text{train}}, D_{\text{val}}, \delta, f^t, \varepsilon)$ (Alg. 2), and normalize $A^t$ to get $\bar{A}^t$.
6:     $p_j^t \propto p_j^{t-1} \exp(\eta \sum_{i=1}^m \bar{A}_{ij}^t)$ for all $j \in [m]$.
7:     Train model $f^{t+\delta}$ with $\frac{S}{T}(1-\delta)$ steps from mixture $p^t$ over $D_{\text{train}}$. Obtain updated $f^{t+1}$.
8: **end for**

---

**Algorithm 2** LEARNPARAMS

1: **Input:** $D_{\text{train}}, D_{\text{val}}$, $\delta$, model $f^t$, number of sweeps $k$, one-hot smoothing factor $\varepsilon$.
2: Split the fraction of a training round $\delta$ into $K$ intervals, where $K = mk$.
3: Set $\beta = 0_{m,m}$.
4: Define $p^{t,i} = (1-\varepsilon)\mathbf{1}_i + \varepsilon \text{Unif}(m)$ for $i \in [m]$, and define $P = [p^{t,1},...,p^{t,m}] \in \triangle^{m \times m}$
5: Randomly shuffle $k$ instances of each $i \in [m]$ to create an order $\mathcal{I} \in [m]^K$.
6: **for** $\tau = 1,...,K$ **do**
7:     Let $j = \mathcal{I}_\tau$. Train model on mixture $p^{t,j}$ of $D_{\text{train}}$ for one interval, obtain $f^{t+\tau\delta/K}$.
8:     **for** $i \in [m]$ **do**
9:         Update $\beta_{ij} \leftarrow \beta_{ij} + L_{\text{val},i}(f^{t+(\tau-1)\delta/K}) - L_{\text{val},i}(f^{t+\tau\delta/K})$ with loss difference on $D_{\text{val}}^i$.
10:     **end for**
11: **end for**
12: Update $\beta \leftarrow \frac{\beta}{k}$.
13: Set $A_i^t = P^{-1}\beta_i$ for each $i \in [m]$.
14: **Return** $A^t \in \mathbb{R}^{m \times m}, f^{t+\delta}$

---

## 6 EXPERIMENTAL RESULTS

We evaluate all methods in the LMO framework, including AIOLI, in two settings. First, we consider an **unrestricted** additional training budget setting to assess how AIOLI compares to other methods in their original form, since each method uses a different number of extra training runs to learn proportions

Table 2: Difference in average test perplexity compared to stratified sampling in the unrestricted setting, where all methods can use $\leq 10$ extra runs to learn $p$. Negative values (green) = improvement. A=Arxiv, B=Books, GH=GitHub, SE=StackExchange, W=Wikipedia.

| Method | A/SE | GH/C4 | B/SE | A/B/SE | CC/GH/W | SlimPajama | # < stratified | # extra runs |
|---|---|---|---|---|---|---|---|---|
| Stratified | 16.532 | 35.991 | 47.192 | 35.114 | 41.583 | 26.426 | - | 0 |
| GS | $-0.399$ | $-0.407$ | $-0.645$ | $-0.247$ | 0.298 | 0.490 | 4 | 10 |
| DML | $-0.241$ | $-0.110$ | $-0.644$ | $-0.599$ | 0.242 | 1.641 | 4 | 10 |
| Skill-It | $-0.326$ | 0.551 | $-0.728$ | $-0.568$ | $-0.195$ | $-0.184$ | 5 | $m$ |
| DoReMi | $-0.307$ | 5.303 | $-0.217$ | $-0.393$ | 6.898 | 0.703 | 3 | 2 |
| DoGE | 0.419 | 0.184 | $-0.678$ | 1.843 | 0.604 | 0.949 | 1 | 1 |
| AIOLI | $-0.205$ | $-0.340$ | $-0.439$ | $-0.226$ | $-0.196$ | $-0.240$ | 6 | 0 |

(Section 6.1). Second, we consider a **restricted** training budget setting to assess if AIOLI can enhance existing methods in practical, budget-constrained conditions, where existing methods have less than a full training run to learn mixing proportions (Section 6.2). Hyperparameters and experimental details, including proportion trajectories are available in Appendix E. Downstream evaluation, ablations, experiments on larger models, and results adapting AIOLI to an out-of-domain setting are in Appendix F.

**Data settings and models.** We use the same data settings and models as in Section 4.1, where we train for $S = 5$K steps for $m = 2,3$-group settings and $S = 40$K steps for the full SlimPajama.

**Baselines and evaluation.** We consider three online methods (Skill-It, DoGE, DoReMi) and one offline method (DML). We also consider grid search (GS), which sweeps training runs and selects $p$ with the lowest average validation loss, and stratified sampling, which sets $p_i = \frac{1}{m}$ for all $i \in [m]$. For each method, we report the average test perplexity per group of the trained model. This metric is considered a proxy for downstream performance (Fan et al., 2024) and also represents the objective in the data mixing problem.

## 6.1 UNRESTRICTED SETTING

**Setup.** We allow methods up to $10S$ additional training steps to learn the mixture proportions. Approaches like grid search and DML can use the entire budget (searching and fitting over 10 full runs), while Skill-It, DoReMi, and DoGE use $mS$, $2S$, and $S$ extra training steps, respectively (see Section 3.3). Stratified sampling and AIOLI use no extra training steps. We evaluate AIOLI with $S_{\text{init}} = 0$.

**Results.** In Table 2, we find that AIOLI robustly outperforms stratified sampling in all 6 data settings by an average of 0.274 perplexity points, while all other methods do worse than stratified sampling on at least 1 set of data groups by up to 6.9 points. The performance of AIOLI and other online methods is additionally reflected in Figure 3, in which we find that AIOLI's $A^t$ similarity with $A^{t\star}$ is correlated with performance. While AIOLI's parameter similarity is not always the highest, we note that its lowest similarity score is much higher than that of other methods, providing evidence that AIOLI's parameter estimation procedure is more consistently accurate than that of other methods. Lastly, regarding offline methods, we hypothesize that their poor performance on settings with larger $m$ is due to the training budget being limited to $10S$, and that increasing this budget would eventually allow them to perform well.

## 6.2 RESTRICTED SETTING

**Motivation.** We introduce the restricted setting because practitioners may not have the resources or desire to complete multiple full training runs, especially as recent LLMs are trained for longer and on more data (Muennighoff et al., 2023). As a result, practitioners may only use data mixing methods on shortened runs, producing learned proportions that may be suboptimal on the full run. We study if AIOLI is able to improve performance by dynamically adjusting previously learned proportions throughout the full training run.

**Setup.** We allow all existing methods up to $0.5S$ additional training steps to learn the mixture proportions. This requires methods to learn $p^{\text{method}}$ over shorter runs of $S_{\text{method}}$ steps each. For instance, grid search will conduct 10 runs of length $S/20$ (see Table 9). We evaluate each method by using $p^{\text{method}}$ learned from shorter runs to train the model on the full run of $S$ steps. We use AIOLI to dynamically adjust each $p^{\text{method}}$ throughout the full run. That is, for each existing method, we run AIOLI with $p^{\text{init}} = p^{\text{method}}$ and $S_{\text{init}} = S_{\text{method}}$, referring to this as AIOLI +method.

**Results.** In Table 3, we find that adding AIOLI to any existing method that learns proportions over shorter runs improves average test perplexity per group in 28 out of 30 settings, by an average of 1.202 and a

Table 3: Average test perplexity in the restricted setting, where each method learns $p$ on shortened runs, and AIOLI +method dynamically adjusts $p$ throughout training. green=AIOLI +method outperforms method.

| Method | Arxiv/SE | GH/C4 | Books/SE | Arxiv/Books/SE | CC/GH/Wiki | SlimPajama |
|---|---|---|---|---|---|---|
| GS | 16.573 | 36.345 | 47.063 | 35.174 | 42.767 | 27.741 |
| AIOLI + GS | 16.388 | 35.925 | 46.667 | 34.705 | 41.378 | 25.654 |
| DML | 16.659 | 36.658 | 46.846 | 34.585 | 42.731 | 37.696 |
| AIOLI + DML | 16.277 | 35.856 | 46.710 | 34.529 | 41.595 | 25.654 |
| Skill-it | 16.246 | 37.255 | 46.667 | 34.539 | 42.069 | 26.734 |
| AIOLI + Skill-it | 16.261 | 36.153 | 46.586 | 34.565 | 41.732 | 26.073 |
| DoReMi | 16.522 | 37.812 | 46.489 | 34.934 | 42.738 | 28.762 |
| AIOLI + DoReMi | 16.347 | 35.626 | 46.163 | 34.770 | 41.800 | 26.587 |
| DoGE | 16.853 | 35.795 | 46.743 | 35.775 | 41.790 | 32.301 |
| AIOLI + DoGE | 16.473 | 35.632 | 46.145 | 34.771 | 41.378 | 26.073 |

maximum of 12.012 points. Furthermore, AIOLI can help methods that initially underperform stratified sampling surpass it, such as DoGE across all settings. In some settings, such as Books/StackExchange, AIOLI improves methods that already outperform stratified sampling. This shows that AIOLI can enhance a wide variety of static proportions, regardless of their initial performance. For the two settings where AIOLI underperforms the base method, the base method already outperforms stratified, and adding AIOLI maintains this trend, worsening perplexity by at most 0.025 points.

## 7 RELATED WORK

**Data mixing.** Albalak et al. (2023) frames online data mixing as a multi-armed bandit problem. Recent works have also studied how to generalize data mixes from smaller to larger models (Kang et al., 2024; Ge et al., 2024; Liu et al., 2024; Allal et al., 2025). Thrush et al. (2024) optimizes model performance on downstream tasks by constructing an $A^t$-like data interactions matrix using pre-trained model perplexities. Jiang et al. (2025) mixes data online using each data group's loss, effectively using a diagonal $A^t$ matrix.

**Curriculum Learning.** Bengio et al. (2009) initially introduced curriculum learning as training models over samples from easiest to hardest. While early work focused on manually designed curricula, later work emphasizes model-driven ones (Hacohen & Weinshall, 2019; Varshney et al., 2022; Fan & Jaggi, 2023; Mindermann et al., 2022). Curricula can encourage skills-based generalization (Huang et al., 2024), or emphasize high quality data to improve downstream task performance (Blakeney et al., 2024).

**Data Selection.** A common way to curate datasets besides mixing is to select data at the per-sample level (Albalak et al., 2024). Techniques here can be broadly classified as data filtering, data matching, and data condensation. In data filtering, low-quality samples are removed using simple heuristics, such as GitHub file lengths (Together.ai, 2023; Touvron et al., 2023), or via deduplication (Abbas et al., 2023; Tirumala et al., 2023; Lee et al., 2022). In data matching, samples that are most similar to a reference dataset are selected. Similarity can be defined in terms of embeddings Xie et al. (2023c), gradients (Xia et al., 2024; Engstrom et al., 2024), or directly using machine learning models to score samples (Brown et al., 2020; Grave et al., 2018; Moore & Lewis, 2010). Lastly, data condensation aims to identify a subset of samples that captures the full training dataset's properties. Selection mechanisms include using gradients, model predictions, and embedding distances (Toneva et al., 2019; Paul et al., 2021; Sorscher et al., 2023).

## 8 DISCUSSION

We introduce the LMO framework, which unifies existing data mixing methods by viewing them as solutions to a common optimization problem involving an implicit method-dependent mixing law. Using this framework, we find that existing methods perform poorly on some datasets due to inaccurate mixing law parameters. This insight inspires AIOLI, whose performance gains are rooted in its ability to estimate parameters $A^t$ of the linear dynamic mixing law throughout training.

**Limitations and Future Work** AIOLI incurs extra inference cost via the repeated evaluations in LEARN-PARAMS (Alg. 2). This can be reduced by computing $L_{\text{val}}$ over a subset of $D_{\text{val}}$, and by using each $A^t$ for longer (decreasing $T$). Another direction is understanding the role of data group partitions. For example, C4 is a subset of CommonCrawl, and it is unclear if disjoint groups could improve performance.

The LMO framework itself is an invitation for future work. It shows that data mixing methods can be improved and analyzed by studying their assumptions on how models learn from data. By exposing such assumptions, LMO identifies key axes for improvement (mixing law parameterization, parameter estimation, and how to solve for $p$), which we hope will inspire new principled data mixing methods.

# 9 ACKNOWLEDGMENTS

We thank Sabri Eyuboglu, Neel Guha, Ben Viggiano, Dan Biderman, Dan Fu, Michael Wornow, Jon Saad-Falcon, Alyssa Unell, Owen Dugan, Jerry Liu, and Gautam Machiraju for their feedback. We thank NYU HPC and Stanford NLP for providing compute and research support. This research project has benefited from the Microsoft Accelerate Foundation Models Research (AFMR) grant program.

We gratefully acknowledge the support of NIH under No. U54EB020405 (Mobilize), NSF under Nos. CCF2247015 (Hardware-Aware), CCF1763315 (Beyond Sparsity), CCF1563078 (Volume to Velocity), and 1937301 (RTML); US DEVCOM ARL under Nos. W911NF-23-2-0184 (Long-context) and W911NF-21-2-0251 (Interactive Human-AI Teaming); ONR under Nos. N000142312633 (Deep Signal Processing); Stanford HAI under No. 247183; NXP, Xilinx, LETI-CEA, Intel, IBM, Microsoft, NEC, Toshiba, TSMC, ARM, Hitachi, BASF, Accenture, Ericsson, Qualcomm, Analog Devices, Google Cloud, Salesforce, Total, the HAI-GCP Cloud Credits for Research program, the Stanford Data Science Initiative (SDSI), the Samsung Advanced Institute of Technology (under the project Next Generation Deep Learning: From Pattern Recognition to AI), the NSF Graduate Research Fellowship (MYH), and members of the Stanford DAWN project: Meta, Google, and VMWare. The U.S. Government is authorized to reproduce and distribute reprints for Governmental purposes notwithstanding any copyright notation thereon. Any opinions, findings, and conclusions or recommendations expressed in this material are those of the authors and do not necessarily reflect the views, policies, or endorsements, either expressed or implied, of NIH, ONR, or the U.S. Government.

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

## APPENDIX

In Appendix A, we provide a glossary of notation used in the paper. In Appendix B, we discuss how additional data mixing methods are related to the LMO framework and provide proofs that existing methods can be expressed in our framework. In Appendix C, we provide additional results on our analysis of existing data mixing methods. In Appendix E we provide additional details for our results in Section 6, and in Appendix F we provide additional results, including downstream evaluation and ablations.

## A  NOTATION

The glossary is given in Table 4 below.

| Symbol | Used for |
|---|---|
| $m$ | The number of data groups. Examples of data groups include a pre-training domain or an instruction-tuning task. |
| $D_{\text{train/val/test}}$ | Training, validation, and test datasets comprised of $m$ groups, where $D_{(\cdot)}^i$ is group $i$'s training/validation/test data. |
| $N$ | Total number of samples from $D_{\text{train}}$ to train on. |
| $S$ | Number of steps to train for (i.e., $S = N \times$ batch size). |
| $T$ | Number of rounds to divide training into, where each round is $\frac{S}{T}$ steps. |
| $\boldsymbol{p}$ | Mixture proportions are $\boldsymbol{p} = (p^1)$ for $T = 1$ (static) and $\boldsymbol{p} = (p^1, ..., p^T)$ for $T > 1$ (dynamic), where $p^t = [p_1^t, ..., p_m^t] \in \triangle^m$ is a probability distribution. |
| $f$ | A language model (can be either pre-trained or initialized from scratch). |
| $f(\boldsymbol{p}, t)$ | The model $f$ at the beginning of round $t$ after being trained on $p^1, ..., p^{t-1}$ so far. |
| $L_{\text{train/val/test}}(f)$ | $L_{\text{train}}(f) = (L_{\text{train},1}(f), ..., L_{\text{train},m}(f))$ is the vector of $f$'s training losses over each data group; similarly defined for validation and test losses. |
| $L_{(\cdot)}^t(\boldsymbol{p})$ | Shorthand for $L_{(\cdot)}(f(\boldsymbol{p}, t))$. When dealing with static mixtures, we also use $L_{(\cdot)}(\boldsymbol{p})$. |
| $A^t$ | Parameter matrix $A^t \in \mathbb{R}^{m \times m}$ used in mixing laws (2), capturing cross-group interactions. See Table 1 for instantiations. |
| $b^t, c^t$ | Group-specific parameters $b^t, c^t \in \mathbb{R}^m$ used in mixing laws 2. Note that the value of $c^t$ does not impact the LMO framework, and neither does $b^t$ when all $b_i^t$ are equal. |
| $\sigma$ | Either $\sigma : \mathbb{R} \to \mathbb{R} = \text{Id}$ or $\exp$. |
| $Z^t$ | Used for normalization in proportion update rule. |
| $\eta$ | Step size $\eta > 0$ used in proportion update rule. |
| $\mathcal{P}$ | The set of mixture proportions that comprises a training sweep. |
| $A^{t\star}$ | Approximately optimal $A^t$ for the linear dynamic mixing law, obtained by fitting $L\text{val}^{t+1}(\boldsymbol{p}) = L\text{val}^t(\boldsymbol{p}) - A^{t\star}\boldsymbol{p}$ over training sweeps. |
| $\tilde{A}^t$ | Method-specific $\tilde{A}^t = b^t A^t$, where $A^t$ is obtained directly from the method and $b^t \in \mathbb{R}$ is learned from training sweeps. |
| $\text{sim}(\tilde{A}^t, A^{t\star})$ | Similarity between method-specific and optimal $A^t$, defined as an average of cosine similarity and Spearman rank correlation over $A^t$'s normalized column sums. |
| $\varepsilon$ | one-hot smoothing factor used to define $p^{t,i} = (1 - \varepsilon)\mathbf{1}_i + \varepsilon\text{Unif}(m)$, smoothed one-hot distributions we use to learn $A^t$ in AIOLI. |
| $\delta$ | The fraction per round dedicated to learning $A^t$ in AIOLI. |
| $k$ | Number of sweeps per group to average $A^t$ estimates over in AIOLI. |
| $\boldsymbol{p}^{\text{init}}$ | Initial mixture $\boldsymbol{p}^{\text{init}} \in \triangle^m$ that AIOLI can dynamically adjust. |
| $S_{\text{init}}$ | Number of steps to train according to $\boldsymbol{p}^{\text{init}}$. |

Table 4: Glossary of variables and symbols used in this paper.

## B  LMO FRAMEWORK DETAILS

### B.1  ADDITIONAL EXISTING METHODS

We comment on two other popular data mixing methods, Online Data Mixing (ODM) (Albalak et al., 2023) and RegMix (Liu et al., 2024).

In ODM (Albalak et al., 2023), data mixing is framed as a multi-armed bandit problem, where each arm is a data group that a batch is trained on, and the reward function is defined in terms of the training

loss of each group. ODM uses the EXP3 algorithm to explore training on different data groups. $p^t$, which is used to determine which group the entire training batch is comprised of, is updated according to $p_j^{t+1} = (1 - m\varepsilon_t)\frac{\exp(\varepsilon_{t-1}R_j^t)}{\sum_{i=1}^m \exp(\varepsilon_{t-1}R_i^t)} + \varepsilon_t$. $\varepsilon_t$ is an exploration rate, and the reward function is $R_j^t = \alpha R_j^{t-1} + (1-\alpha)\frac{L_{\text{train},j}^t(\boldsymbol{p})}{p_j^t}$ if the $j$th group is selected at time $t$; otherwise, $R_j^t = R_j^{t-1}$. While the exploration and the smoothing of $p^t$ and $R^t$ make this method not directly expressible in our framework, we note that the update rule can be loosely interpreted as allocating larger proportions to groups that have high loss. This update rule does not consider cross-group interactions and is thus similar to DoReMi's update rule, which utilizes a diagonal $A^t$ defined in terms of current loss.

RegMix (Liu et al., 2024) conducts many training runs on smaller models at shorter scales. Similar to DML (Ye et al., 2024), a regression model is fit to these runs and used to predict mixture proportions for a longer run on a larger model. They consider using a linear regression model, i.e., the mixing law $L_{\text{val},i}(\boldsymbol{p}) = c_i - \sum_{j=1}^m A_{ij}p_j^t$, but find that the $R^2$ is relatively low (0.87). Instead, their main approach uses LightGBM, a tree-based gradient boosting approach, i.e., using an ensemble of non-linear decision trees as a mixing law. We note that AIOLI could be used in conjunction with RegMix in their settings, an exciting direction for future work.

## B.2 Proofs for section 3.3

### B.2.1 Background on Exponentiated Gradient Descent

We provide background on exponentiated gradient descent (EGD) taken from Kakade (n.d.). In EGD, we have a sequence of decisions $w^1, ..., w^T$, where $w^t = [w_1^t, ..., w_m^t] \in \triangle^m$. We also have a sequence of cost functions $c^1, ..., c^T : \triangle^m \to \mathbb{R}$. To minimize the total cost $\sum_{t=1}^T c^t(w^t)$, the EGD update rule sets $w^0 = \text{Unif}(m)$, and updates according to $w_j^{t+1} = \frac{w_j^t \exp(-\eta \nabla_j c^t(w^t))}{Z_t}$. $Z_t$ ensures that $w^{t+1} \in \triangle^m$, $\eta$ is a step size, and $\nabla_j c^t(w^t)$ denotes $\frac{\partial c^t(w^t)}{\partial w_j^t}$. EGD is known to have certain regret guarantees on the value of costs incurred by playing $w^1, ..., w^T$ versus always playing the best fixed point in hindsight: $\sum_{t=1}^T c^t(w^t) - \inf_{w \in \triangle^m} \sum_{t=1}^T c^t(w)$.

We now are ready to prove Lemma 1.

**Lemma 1.** *The EGD update rule for* (1) *subject to* $L_{\text{val},i}^{t+1}(\boldsymbol{p}) = c_i^t - b_i^t \sum_{j=1}^m A_{ij}^t p_j^t \ \forall i \in [m]$ *is*

$$p_j^{t+1} = \frac{1}{Z^t} \cdot p_j^t \exp\left(\eta \sum_{i=1}^m b_i^t A_{ij}^t\right) \forall j \in [m], \tag{3}$$

*where $\eta > 0$ is the step size and $Z^t$ is a normalizing constant such that $p_j^{t+1} \in \triangle^m$.*

*Proof.* The cost function at each timestep in our setting is $\sum_{i=1}^m L_{\text{val},i}^{t+1}(\boldsymbol{p})$, and the decision we make is $p^t$. The mixing law constraint in (2) with $\sigma = \text{Id}$ is $L_{\text{val},i}^{t+1}(\boldsymbol{p}) = c_i^t - b_i^t \sum_{j=1}^m A_{ij}^t p_j^t$ for all $i \in [m]$, so our objective (1) can be written as

$$\sum_{i=1}^m \left(c_i^t - b_i^t \sum_{j=1}^m A_{ij}^t p_j^t\right). \tag{4}$$

The gradient of this expression with respect to $p_j^t$ for $j \in [m]$ is $-\sum_{i=1}^m b_i^t A_{ij}^t$. Plugging this into the EGD update rule, we obtain the update $p_j^{t+1} = \frac{1}{Z_t} p_j^t \exp(\eta \sum_{i=1}^m b_i^t A_{ij}^t)$. $\square$

### B.2.2 Proof of Theorem 1

To prove Theorem 1, we write out individual propositions 1, 2, 3 for expressing each online method in the LMO framework.

By our definition of what it means to express a method in LMO, we must consider how each method 1) trains $f$ and 2) sets $p^t$. We must see if this procedure can be replicated by solving some specification of the LMO optimization problem in our data mixing setup.

Critically, note that this definition of "expression" does not claim that the optimization problems proposed in existing methods are exactly the same as the LMO optimization problem. Instead, we are stating that the training procedures used in their methods can be equivalently viewed as a way of solving the LMO optimization problem subject to certain assumptions on the loss-proportion relationship.

**Proposition 1** (Skill-It Derivation). *Using* a) *a linear dynamic parameterization* $L_{val,i}^{t+1}(\boldsymbol{p}) = L_{val,i}^{t}(\boldsymbol{p}) - b^t \sum_{j=1}^{m} A_{ij}^t p_j^t$, b) *parameters* $A_{ij}^t = L_{val,i}^t(\boldsymbol{p}) \cdot (L_{val,i}^{T+1}(\mathbf{1}_j) - L_{val,i}^1(\mathbf{1}_j))/L_{val,i}^1(\mathbf{1}_j)$, *and* c) *exponentiated gradient descent (EGD) to solve for* $\boldsymbol{p}$, *the LMO framework* (1) *can express Skill-It.*

*Proof.* The Skill-It algorithm sets $p^t$ in each round and then samples from $D_{\text{train}}$ according to $p^t$ to train $f$ for a round. This training procedure is directly specified in our data mixing problem setup (Section 2). Therefore, we simply need to show that the Skill-It update rule can be converted into a linear dynamic mixing law. By comparing Lemma 1 and the Skill-It update rule $p_j^{t+1} = \frac{1}{Z_t} \cdot p_j^t \exp\left(\eta \sum_{i=1}^{m} A_{ij}^{\text{SG}} L_{\text{val},i}^t(\boldsymbol{p})\right)$, we can match $A_{ij}^t$ in the lemma with $A_{ij}^{\text{SG}}$ in Skill-It, and we can match $b_i^t$ in the lemma with $L_{\text{val},i}^t(\boldsymbol{p})$. Therefore, Lemma 1 tells us that using $L_{\text{val},i}^{t+1}(\boldsymbol{p}) = c_i^t - b^t \sum_{j=1}^{m} L_{\text{val},i}^t(\boldsymbol{p}) A_{ij}^{\text{SG}} p_j^t$ in the LMO framework with exponentiated gradient descent recovers Skill-It (since the $b^t$ and $c_i^t$ can be dropped and are only used for scaling $A^t$).

Using the definition of $A_{ij}^{\text{SG}}$, we can rewrite the mixing law as $L_{\text{val},i}^{t+1}(\boldsymbol{p}) = c_i^t - b^t \sum_{j=1}^{m} A_{ij}^{t,\text{Skill-It}} p_j^t$ where $A_{ij}^{t,\text{Skill-It}} = L_{\text{val},i}^t(\boldsymbol{p})(L_{\text{val},i}^{T+1}(\mathbf{1}_j) - L_{\text{val},i}^1(\mathbf{1}_j))/L_{\text{val},i}^1(\mathbf{1}_j)$. Lastly, note that we can replace $c_i^t$ with any other value, including $L_{\text{val},i}^t(\boldsymbol{p})$, due to the fact that $p^t$ has $m-1$ degrees of freedom (see Lemma 2).

We note that Chen et al. (2023) explicitly specify their mixing law in equation 2 of their paper, along with the same objective function as ours in the LMO framework. □

**Proposition 2** (DoReMi Derivation). *Using* a) *a linear dynamic parameterization* $L_{val,i}^{t+1}(\boldsymbol{p}) = L_{val,i}^{t}(\boldsymbol{p}) - b^t \sum_{j=1}^{m} A_{ij}^t p_j^t$, b) *parameters* $A_{ij}^t = \min\{L_{train,i}^t(\boldsymbol{p}) - L_{train,i}(f_{ref}), 0\}$ *for* $i=j$ *and* $A_{ij}=0$ *otherwise, and* c) *EGD to solve for* $\boldsymbol{p}$, *the LMO framework* (1) *can express DoReMi's proxy model.*

*Proof.* When training the proxy model for DoReMi, $p^t$ is set in each round, and then $f$ is updated to minimize $\sum_{i=1}^{m} p_i^t L_{\text{train},i}(f)$. Using Lemma 3, we establish that DoReMi's weighted training objective at each timestep is equal in expectation to the objective of training on data sampled from $p^t$, which is what our problem setup focuses on. Having established that the training procedure is the same in expectation, we now need to show that the DoReMi $p^t$ update rule can be converted into a linear dynamic mixing law. By comparing Lemma 1 and the DoReMi update rule $p_j^{t+1} \propto p_j^t \exp(\eta \max\{L_{\text{train},j}^t(\boldsymbol{p}) - L_{\text{train},j}(f_{\text{ref}}), 0\})$, we can match $A_{ij}^t$ in the lemma with 0 for $i \neq j$, and $A_{ii}^t$ with $\max\{L_{\text{train},j}^t(\boldsymbol{p}) - L_{\text{train},j}(f_{\text{ref}}), 0\}$. Therefore, Lemma 1 tells us that using $L_{\text{val},i}^{t+1} = c_i^t - b^t \sum_{j=1}^{m} A_{ij}^t p_j^t$ with $A_{ii}^t = \max\{L_{\text{train},j}^t(\boldsymbol{p}) - L_{\text{train},j}(f_{\text{ref}}), 0\}$ can express the DoReMi proxy model training. We include $b^t$ to allow for scaling $A^t$, but since this does not impact the optimal $\boldsymbol{p}$, it is not in the update rule. Lastly, applying Lemma 2 lets us write the mixing law as $L_{\text{val},i}^{t+1} = L_{\text{val},i}^t(\boldsymbol{p}) - b^t \sum_{j=1}^{m} A_{ij}^t p_j^t$.

We comment on the fact that DoReMi's proxy model is trained with a DRO (distributionally robust optimization) min-max objective, namely, $\text{minimize}_f \text{maximize}_p \sum_{i=1}^{m} p_i L_{\text{train},i}^{T+1}(f)$. This objective, which differs from our data mixing objective, yields the $p^t$ gradient ascent and $f^t$ gradient descent updates. However, we are still able to express this training procedure in the LMO framework, since our claim is: if we assume that the $L_{\text{val},i}^{t+1} = L_{\text{val},i}^t(\boldsymbol{p}) - b^t \sum_{j=1}^{m} A_{ij}^{t,\text{DRM}} p_j^t$ mixing law captures the relationship between $L_{\text{val}}^t$ and $p^t$, then training according to the DoReMi proxy run should not only guide $f$ and $\boldsymbol{p}$ to optimize the DRO objective, but also to optimize the average validation loss per group.

□

**Proposition 3** (DoGE Derivation). *Using* a) *a linear dynamic parameterization* $L_{val,i}^{t+1}(\boldsymbol{p}) = L_{val,i}^{t}(\boldsymbol{p}) - b^t \sum_{j=1}^{m} A_{ij}^t p_j^t$, b) *parameters* $A_{ij}^t = \langle \nabla L_{val,i}^t(\boldsymbol{p}), \nabla L_{train,j}^t(\boldsymbol{p}) \rangle$ *for all* $i,j \in [m]$, *and* c) *EGD to solve for* $\boldsymbol{p}$, *the LMO framework* (1) *can express DoGE's proxy model.*

*Proof.* When training the proxy model for DoGE, $p^t$ is set in each round, and then $f$ updated to minimize $\sum_{i=1}^{m} p_i^t L_{\text{train},i}(f)$. Using Lemma 3, we establish that DoGE's weighted training objective at each timestep

is equal in expectation to the objective of training on data sampled from $p^t$. Next, we show that the DoGE update rule can be converted into a linear dynamic mixing law. By comparing Lemma 1 and the DoGE update rule $p_j^{t+1} \propto p_j^t \exp(\eta \langle \nabla L_{\text{train},j}(f^t), \sum_{i=1}^m \nabla L_{\text{val},i}(f^t) \rangle)$, we can see that $A_{ij}^t$ in the Lemma can be matched with $\langle \nabla L_{\text{train},j}(f^t), \nabla L_{\text{val},i}(f^t) \rangle$. Therefore, using the mixing law $L_{\text{val},i}^{t+1} = c_i^t - b^t \sum_{j=1}^m A_{ij}^t p_j^t$ with $A_{ij}^t = \langle \nabla L_{\text{train},j}(f^t), \nabla L_{\text{val},i}(f^t) \rangle$ allows LMO to express DoGE proxy model training. Again, $b^t$ is included for scaling but does not impact optimization, and by applying Lemma 2, we can replace $c_i^t$ with $L_{\text{val},i}^t(\boldsymbol{p})$. $\square$

**Lemma 2.** *Let* $L_i^{t+1}(\boldsymbol{p}) = c_i^t - \sum_{j=1}^m A_{ij}^t p_j^t$ *for some* $c^t$ *and* $A^t$. *Then, there exists an* $B_{ij}^t$ *such that* $L_i^{t+1}(\boldsymbol{p}) = L_i^t(\boldsymbol{p}) - \sum_{j=1}^m B_{ij}^t p_j^t$.

*Proof.* Since $p^t \in \triangle^m$, we can write the probability $p_m^t$ as $1 - \sum_{j=1}^{m-1} p_j^t$. Then, the first equation can be written as

$$L_i^{t+1}(\boldsymbol{p}) = c_i^t - \sum_{j=1}^{m-1} A_{ij}^t p_j^t - A_{im}^t \left(1 - \sum_{j=1}^{m-1} p_j^t\right) \tag{5}$$

$$= c_i^t - \sum_{j=1}^{m-1} (A_{ij}^t - A_{im}^t) p_j^t - A_{im}^t$$

$$= L_i^t(\boldsymbol{p}) - \sum_{j=1}^{m-1} (A_{ij}^t - A_{im}^t) p_j^t - (A_{im}^t - c_i^t + L_i^t(\boldsymbol{p}))$$

$$= L_i^t(\boldsymbol{p}) - \sum_{j=1}^{m-1} (A_{ij}^t - A_{im}^t + A_{im}^t - c_i^t + L_i^t(\boldsymbol{p})) p_j^t - (A_{im}^t - c_i^t + L_i^t(\boldsymbol{p}))(1 - \sum_{j=1}^{m-1} p_j^t)$$

$$= L_i^t(\boldsymbol{p}) - \sum_{j=1}^{m-1} (A_{ij}^t - c_i^t + L_i^t(\boldsymbol{p})) p_j^t - (A_{im}^t - c_i^t + L_i^t(\boldsymbol{p}))(1 - \sum_{j=1}^{m-1} p_j^t).$$

Let $B_{ij}^t = A_{ij}^t - c_i^t + L_i^t(\boldsymbol{p})$ for all $j \in [m]$. Then, this equation becomes

$$L_i^{t+1}(\boldsymbol{p}) = L_i^t(\boldsymbol{p}) - \sum_{j=1}^{m-1} B_{ij}^t p_j^t - B_{im}^t (1 - \sum_{j=1}^{m-1} p_j^t) \tag{6}$$

$$= L_i^t(\boldsymbol{p}) - \sum_{j=1}^m B_{ij}^t p_j^t.$$

$\square$

**Lemma 3.** *Let* $L_B^t(f,p)$ *be the total training loss of* $f$ *on a batch of size* $B$ *sampled from* $D_{train}$ *according to* $p \in \triangle^m$, *and let* $L_{B,i}^t(f,p)$ *be the total training loss on samples from group* $i$ *in that batch. Then, the average loss over a uniformly sampled batch weighted by* $p^t$ *is equal in expectation to the average loss per group over a batch sampled according to* $p^t$:

$$\mathbb{E}\left[\sum_{i=1}^m p_i^t L_{B,i}^t(f, Unif(m))\right] = \mathbb{E}\left[\frac{L_B^t(f, p^t)}{m}\right] \tag{7}$$

*Proof.* Let each group $i$ consist of samples $x$ from the distribution $\mathcal{P}_i$, and let $\tilde{L}_{\text{train, i}}(f) = \mathbb{E}_{x \sim \mathcal{P}_i}[\ell(f,x)]$ be the population-level loss on group $i$, where $\ell(f,x)$ is $f$'s loss on sample $x$.

If a batch is uniformly sampled, each group has $B/m$ samples. We can then write $L_{B,i}^t(f, \text{Unif}(m)) = \sum_{k=1}^{B/m} \ell(f, x_k^i)$, where $x_k^i$ is the $k$th sample of group $i$. Then,

$$\mathbb{E}\left[\sum_{i=1}^m p_i^t L_{B,i}^t(f, \text{Unif}(m))\right] = \mathbb{E}\left[\sum_{i=1}^m p_i^t \sum_{k=1}^{B/m} \ell(f, x_k^i)\right] = \sum_{i=1}^m \frac{p_i^t B}{m} \tilde{L}_{\text{train, i}}(f). \tag{8}$$

Next, if a batch is sampled according to $p^t$, then group $i$ has $Bp_i^t$ samples in the batch. We can then write $L_B^t(f,p^t) = \sum_{i=1}^m \sum_{k=1}^{p_i^t B} \ell(f,x_k^i)$. Then,

$$\mathbb{E}\left[\frac{L_B^t(f,p^t)}{m}\right] = \mathbb{E}\left[\sum_{i=1}^m \sum_{k=1}^{p_i^t B} \frac{\ell(f,x_k^i)}{m}\right] = \sum_{i=1}^m \frac{p_i^t B}{m}\tilde{L}_{\text{train, i}}(f). \tag{9}$$

This hence establishes the equivalence in expectation between a weighted training objective and training on data sampled according to $\boldsymbol{p}$.

$\square$

## C  ANALYSIS DETAILS

### C.1  MIXING LAW PARAMETERIZATION

Table 5: Comparison of log-linear static and linear dynamic mixing law parameterizations across different data settings with MSE and $R^2$ metrics. Both log-linear and linear dynamic mixing laws fit the relationship between mixing proportions and losses well.

| Parameterization | Arxiv/SE | | GH/C4 | | Books/SE | |
| --- | --- | --- | --- | --- | --- | --- |
| | MSE | $R^2$ | MSE | $R^2$ | MSE | $R^2$ |
| Log-linear static | 2e-4 | 0.990 | 5e-4 | 0.989 | 6e-4 | 0.987 |
| Linear dynamic | 2e-4 | 0.936 | 1e-4 | 0.948 | 4e-5 | 0.926 |
| | Arxiv/Books/SE | | CC/GH/Wiki | | SlimPajama | |
| | MSE | $R^2$ | MSE | $R^2$ | MSE | $R^2$ |
| Log-linear static | 6e-4 | 0.991 | 0.001 | 0.989 | 0.002 | 0.997 |
| Linear dynamic | 6e-5 | 0.957 | 1e-4 | 0.975 | 5e-6 | 0.938 |

We describe how we performed the linear and log-linear parameterization experiments.

For the log-linear static parameterizations, we train our model on $\boldsymbol{p} \in \mathcal{P}$ sweeps and fit the parameters using code provided in Ye et al. (2024) (i.e., using PyTorch and L-BFGS to minimize the Huber loss of the mixing law). We do this over 5 random seeds for $k=2,3$ and over 3 seeds for the full SlimPajama.

For the linear dynamic parameterizations, for $k=2,3$ we train the model for 2000 steps according to some $p^0 \in \mathcal{P}$, and then sweep over $\mathcal{P}$ for the next 100 steps. We do this for one random seed, performing $|\mathcal{P}|^2$ total runs. For the full SlimPajama setting, we train the model for 10000 steps using stratified sampling, and then sweep over $\mathcal{P}$ for the next 5000 steps. We fit the parameters using Pytorch and L-BFGS.

#### C.1.1  ADDITIONAL PARAMETERIZATION EXPERIMENTS

**Parameterization across checkpoints.**  We investigate whether the log-linear static and linear dynamic mixing laws remain well-specified in later stages of training and on other datasets. To do so, we take various Pythia 160M checkpoints (Biderman et al., 2023), sweep mixing proportions, and fit the linear dynamic and log-linear static mixing laws. We train for 2000 steps according to the learning rates and learning rate scheduler reported in (Biderman et al., 2023). We fit the static mixing law on full runs of 2000 steps, and the linear dynamic mixing law at $t=500$, after which we do a training sweep over the next 500 steps. In Tables 6 and 7, we find that the strong fit for log-linear static mixing laws continues to hold during pre-training at checkpoint 72K (roughly halfway through training Pythia-160M) and after pre-training, with an average $R^2$ of 0.982 and 0.991, respectively. However, the linear dynamic mixing law's $R^2$ coefficient is lower, averaging 0.815 at checkpoint 72K and 0.830 at the end of pre-training. It thus may be interesting to further study if the dynamics of the loss-proportion relationship evolve in a structured way throughout training, or if these results are due to more noise in how models learn at later stages of training.

**Parameterization across other sets of data groups.**  In Figure 4, we identify an example set of data groups that exhibits a *non-linear* relationship between loss and proportion: Books/C4 from SlimPajama.

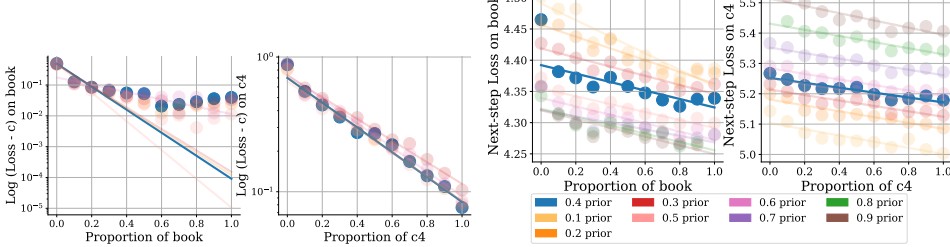

Figure 4: Top: Log-linear static mixing law fit on Books/C4 across 5 random seeds. Bottom: Linear dynamic mixing law fit on Books/C4 on 1 random seed. Each color is a different initial mixture $p^0 \in \mathcal{P}$ trained for 2000 steps, and the fitting sweeps are done over 100 additional steps.

For these two data groups, we see that as the proportion of Books increases while C4 decreases, the loss on Books starts *increasing* past a certain $p$, suggesting quite counterintuitively that performance on Books is optimized by allocating some proportion to C4. In this case, neither log-linear static or linear dynamic mixing laws have good fit to the proportion-loss relationship, as neither can represent the non-linearity. In particular, the average MSE and $R^2$ for the log-linear static mixing law is $0.003$ and $0.558$, respectively, and the average MSE and $R^2$ for the linear dynamic mixing law is $0.0002$ and $0.721$.

Fortunately, because these nonlinearities exist on the boundary of the simplex and tend to incur high loss, they tend to have little impact on the optimization of $p$, which strives to minimize the average loss. For instance, we found that the optimal proportion according to Ye et al. (2024)'s log-linear static mixing law on one random seed was $[0.176, 0.824]$, and the true optimal from grid search was $[0.2, 0.8]$. However, it is important to further investigate this non-linear phenomenon on additional data groups and training regimes, which we defer to future work.

Table 6: Comparison of log-linear static and linear dynamic mixing law parameterizations when training from the 72K Pythia-160M checkpoint.

| Parameterization | Arxiv/SE | | GH/C4 | | Books/SE | |
|---|---|---|---|---|---|---|
| | MSE | $R^2$ | MSE | $R^2$ | MSE | $R^2$ |
| Log-linear static | 2e-4 | 0.975 | 7e-5 | 0.992 | 2e-4 | 0.981 |
| Linear dynamic | 4e-4 | 0.834 | 7e-4 | 0.815 | 6e-4 | 0.796 |

Table 7: Comparison of log-linear static and linear dynamic mixing law parameterizations when training from the pre-trained Pythia-160M.

| Parameterization | Arxiv/SE | | GH/C4 | | Books/SE | |
|---|---|---|---|---|---|---|
| | MSE | $R^2$ | MSE | $R^2$ | MSE | $R^2$ |
| Log-linear static | 3e-6 | 0.994 | 4e-6 | 0.992 | 6e-6 | 0.986 |
| Linear dynamic | 5e-5 | 0.896 | 8e-5 | 0.824 | 1e-4 | 0.769 |

**Checking for interactions among groups.** It is natural to ask whether a *linear* mixing law is sufficient to model how mixing proportions affect the loss. In linear regression, such assumptions are often evaluated using visual diagnostics called *residual plots* (Montgomery et al., 2021). Residual plots graph the prediction error from each data point (the *residuals*) in order to reveal different kinds of structure. For example, it is common to plot the residual against the predicted value to check for nonlinearity.

Figure 5 shows several such residual plots for the dynamic mixing law experiments with 3 domains (Arxiv, Books, and Stackexchange). The figure checks for interactions when predicting Arxiv's loss. The corresponding plots for the other domains look similar.

The top row visualizes the residuals inside the simplex. If strong interactions were present, then they would cause clustered patterns in the residuals—regions where the linear model consistently gives predictions that are too low or too high. Strong patterns do not seem apparent.

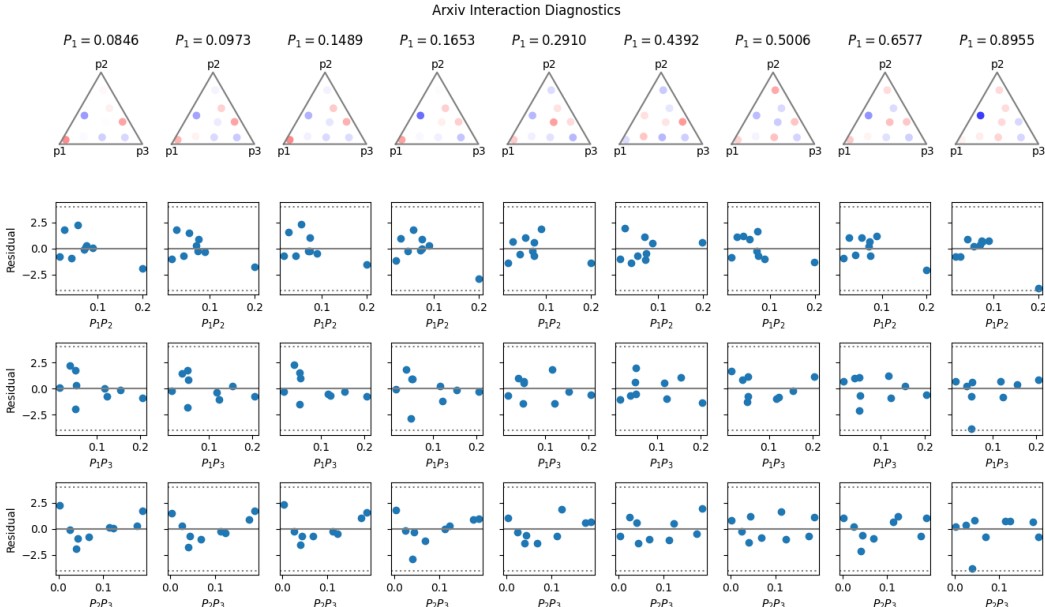

Figure 5: Residuals plots to check for interactions in the dynamic mixing law experiments with 3 domains (Arxiv, Books, and StackExchange). The target loss is Arxiv. Columns correspond to different initial mixing proportions. Data points show the (externally studentized) residuals of different mixing proportions after fitting the linear mixing law. Top row: Each point in the simplex corresponds to a different mixture of the 3 domains, with its color giving the residual's value at that point (red is positive, blue is negative). Bottom 3 rows: each row shows the residual plotted against a different interaction term: $P_1 P_2$, $P_1 P_3$, and $P_2 P_3$. Dotted gray lines show upper and lower 99% confidence limits for the residuals, assuming the linear regression assumptions hold.

The bottom three rows plot the residuals against different interaction terms. A consistent trend in the residuals above or below zero would suggest the term captures a meaningful interaction. The scatter plots show no consistent trend. The first three charts on the bottom row hint that a small interaction could be present in those cases; however, it is difficult to say without larger samples. Considering the linear model's excellent fit and high $R^2$, if such an interaction is present then it is likely small.

To summarize: the linear model seems sufficient. While we can not rule out the possibility of small interactions, the diagnostics do not reveal any major departures from linearity that might compel us to use a more complex model.

## C.2 VALUES OF MIXING LAW PARAMETERS

We explain how to compare method-specific $A^t$'s to an approximation of the true $A^{t\star}$. First, after performing method-specific initialization, such as training reference models, we run each online method (Skill-It, DoReMi's proxy model DoGE's proxy model, Skill-it, and AIOLI) for $t$ steps. For Skill-It, DoReMi, and DoGE, we use the unrestricted setting configuration of hyperparameters presented in Section E. For AIOLI, we analyze the parameters of AIOLI +GS from the restricted setting, since we found that this had less noisy fluctuation in the weights than in the unrestricted setting. For $m = 2$, we set $t = 1000$ for Skill-It and $t = 500$ for DoGE, DoReMi, and AIOLI since Skill-It is updated less frequently. For $m = 3$, we set $t = 1000$ for DoGE, DoReMi and Skill-It, and $t = 1500$ for AIOLI. We then checkpoint the language model and the method's $A^t$. For DoGE and DoReMi, we compute a smoothed $A^t = \frac{1}{100} \sum_{i=1}^{100} A^{t-100+i}$ because each $A^t$ is computed at the batch level, and can thus be noisy. For AIOLI, we also smooth the $A^t$ by averaging the previous timestep parameters.

To approximate $A^{t\star}$, we then run a training sweep of $p^t$ over $\mathcal{P}$ for 100 steps on the checkpoint. We use this training sweep to fit $A^{t\star}$ from the dynamic mixing law $L_{\text{val},i}^{t+1}(\boldsymbol{p}) = L_{\text{val},i}^t(\boldsymbol{p}) - \sum_{j=1}^{m} A_{ij}^{t\star} p_j^t$.

Before we compare parameters, we scale $A^t$ by some $b^t$ where $L_{\text{val},i}^{t+1}(\boldsymbol{p}) = L_{\text{val},i}^t(\boldsymbol{p}) - b^t \sum_{j=1}^m A_{ij}^t p_j^t$ for all $i \in [m]$. This is allowed since $b^t$ does not influence the optimal $\boldsymbol{p}$ and does not need to be in the update rule. We fit a single $b^t$ across each group's mixing law and set $\tilde{A}^t = b^t A^t$. We can then compare $A^t$ and $A^{t\star}$ using the metric $\text{sim}(\tilde{A}^t, A^{t\star}) = 0.5\text{cossim}(\tilde{a}^t, a^{t\star}) + 0.5\text{Spearman}(\tilde{a}^t, a^{t\star})$, which we proposed in Section 4.3.

### C.2.1 PROPERTIES OF $A^{t\star}$

We discuss some properties of $A^{t\star}$, finding that 1) $A^{t\star}$ can vary significantly across time, and 2) $A^{t\star}$ needs to be modeled as a full matrix. To do this, for each initial mixture $p^0 \in \mathcal{P}$, we train for $t = 2000$ steps and then sweep over $\mathcal{P}$ for the next 100 steps. We repeat this setup for $t = 4000$ to obtain $A^{2000\star}$ and $A^{4000\star}$. We do this experiment for Arxiv/Stackexchange and Github/C4.

**Extent of time variation of $A^t$.** We find that the column sums of $A^t$ can change order over time, meaning that the $p^t$ "changes direction" in terms of which group has the largest proportion. In particular, for $p^0 = [0.5, 0.5]$ and Github/C4, we have that

$$A^{2000\star} = \begin{bmatrix} 0.148 & 0.011 \\ -0.013 & 0.087 \end{bmatrix} \qquad A^{4000\star} = \begin{bmatrix} 0.015 & 0.001 \\ 0.001 & 0.015 \end{bmatrix} \tag{10}$$

The column sums are $\mathbf{1}^\top A^{2000\star} = [0.135, 0.098]$ and $\mathbf{1}^\top A^{4000\star} = [0.016, 0.017]$, showing that the ordering of proportions of the groups changes. This suggests that the optimal $p^t$ can change significantly across time, prioritizing Github initially and later C4, which is also reflected for Github/C4 in the greedy row of Table 8.

However, for Arxiv/Stackexchange, the column sums of $A^{2000\star}$ and $A^{4000\star}$ never change in terms of the ordering of proportions of the data groups, across all $p^0 \in \mathcal{P}$. As a result, the optimal $p^t$ never changes direction. This suggests that how much $A^t$ varies in ordering over time depends on the data groups. As a result, methods like Skill-It, which use a time-invariant $A^{\text{SG}}$ multiplied by validation loss, may not be able to match the true $A^{t\star}$ if the groups' validation losses do not change in ranking across time, which we observe in Github/C4.

**Modeling $A^{t\star}$ as a full vs diagonal matrix.** We find that modeling the off-diagonal entries of $A^{t,\star}$ is important. For each sweep, we fit both $A^{t\star}$ as described above and a diagonal matrix $A_d^{t\star}$. We compare if the column sums of $A^{t\star}$ and $A_d^{t\star}$ differ in the order of elements.

We find that for Arxiv/StackExchange, $p^0 = 0.4$, and both $t = 2000$ and $t = 4000$, setting $p^t$ based on the full matrix would put a larger proportion on StackExchange, while setting $p^t$ based on the diagonal matrix would put a larger weight on ArXiv. In particular, the full and diagonal matrices for $t = 2000$ are

$$A^{2000\star} = \begin{bmatrix} 0.249 & 0.058 \\ 0.025 & 0.224 \end{bmatrix} \qquad A_d^{2000\star} = \begin{bmatrix} 0.284 & 0 \\ 0 & 0.238 \end{bmatrix} \tag{11}$$

The second column sum is larger for $A^{2000\star}$ and smaller for $A_d^{2000\star}$. We also have similar findings on Github/C4; for $p^0 = 0.6$ and $t = 2000$, we have

$$A^{2000\star} = \begin{bmatrix} 0.119 & 0.027 \\ -0.010 & 0.104 \end{bmatrix} \qquad A_d^{2000\star} = \begin{bmatrix} 0.135 & 0 \\ 0 & 0.098 \end{bmatrix} \tag{12}$$

Using the diagonal matrix for Github/C4 would result in prioritizing training on Github, even though the full matrix suggests that C4 should be prioritized. Therefore, it is important to model $A^{t\star}$ as a full matrix. As a result, methods like DoReMi, which use a diagonal $A^t$, can perform suboptimally.

### C.3 SOLVING STRATEGY

We present our results on examining the assumptions made in how existing methods solve the LMO optimization problem. All online methods use exponentiated gradient descent, which updates $p^t$ using the gradient at the current timestep. This involves a greedy approximation of the objective function. We study if the greedy approximation yields a $\boldsymbol{p}$ is close to the true optimal $\boldsymbol{p}$.

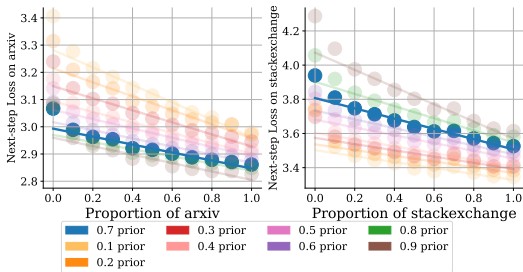

Figure 6: The linear dynamic parameterization results from Figure 2 (right), with $p^t = [0,1]$ and $[1,0]$ also plotted. We see that the linear dynamics are misspecified at $p_i^t = 0$ for both $i$.

For $m = 2$ data settings, we take our $S = 5000$ steps and split it into $T = 2$ rounds. We perform a brute-force sweep at each round over $\mathcal{P}$, which sweeps $p_1 = 0.1, 0.2, ..., 0.9$. In total over one random seed, we conduct 81 training runs for each of Arxiv/Stackexchange, Github/C4, and Books/Stackexchange.

We determine the greedy-approximate $\boldsymbol{p}$ by selecting the best $p^1$. Then, conditioning on this $p^1$, we select the best $p^2$. We report what the greedy $\boldsymbol{p}$ and its performance is in the first row of Table 8, and we report the optimal $\boldsymbol{p}$ and its performance in the second row. Note that this protocol does not depend on the mixing law or a method for setting $\boldsymbol{p}$.

We find that for Arxiv/StackExchange and Books/StackExchange, the greedy proportions and the optimal proportions are identical. However, for Github/C4, the greedy approximation fails to recover the optimal proportions. Therefore, the greedy approximation recovers the optimal dynamic proportions in 2 out of 3 cases.

Table 8: Comparison of the greedily selected $p^1$, $p^2$ versus the optimal $p^1$, $p^2$ for a $T = 2$ rounds data mixing problem. On 2 out of 3 datasets, the greedily selected proportions match the optimal proportions.

| Solving | Arxiv/SE | | GH/C4 | | Books/SE | |
|---|---|---|---|---|---|---|
| | $p_1^1, p_1^2$ | Avg test PPL | $p_1^1, p_1^2$ | Avg test PPL | $p_1^1, p_1^2$ | Avg test PPL |
| Greedy | 0.4, 0.4 | 16.039 | 0.6, 0.4 | 36.525 | 0.3, 0.6 | 45.513 |
| Optimal | 0.4, 0.4 | 16.039 | 0.3, 0.6 | 34.709 | 0.3, 0.6 | 45.513 |

Beyond exponentiated gradient descent, one may wonder if exactly solving the greedy objective could suffice. For the linear dynamic mixing law $L^{t+1}(\boldsymbol{p}) = L^t(\boldsymbol{p}) - A^t p^t$, the optimal $p^t$ is $\mathbf{1}_j$, where $j = \text{argmax} \sum_{i=1}^m A_{ij}^t$. However, we find in Figure 6 that the loss-proportion relationship can be nonlinear at the edge of the simplex where $p^t = \mathbf{1}_j$. Exponentiated gradient descent, which uses entropy regularization, is hence able to implicitly avoid extreme $\boldsymbol{p}$ where the linear mixing law is misspecified and thus is a practical technique for LMO.

## D    ADDITIONAL ALGORITHMIC DETAILS

In AIOLI, LEARNPARAMS is used in each round to learn $A^t$. Then, $A^t$ is used to compute $p^t$, which is used for training during the round. We provide a derivation of LEARNPARAMS by first presenting a naive, high-cost method for estimating $A^t$ (Appendix D.1). This involves checkpointing the model at each round, running a training sweep over the round and observing the changes in validation losses, and fitting $A^t$ to these changes. Then, we layer on two modifications that compute slightly different loss changes, helping lower the cost of estimation. First, we shorten the training sweep to be only over a fraction of the round, $\delta$, and use these shortened changes in validation losses to fit $A^t$ (Appendix D.2). Second, we simulate a simultaneous training sweep by partitioning the $\delta$ fraction of the round into many small parts, interleaving the different sweep mixtures at a fine granularity and averaging the loss changes for each sweep mixture (Appendix D.3). This idea, with similarity to concepts like time-division multiplexing in signal processing (Chaparro & Akan, 2019), enables AIOLI to require no extra training while trading off accuracy of the estimate. We provide a sketch of our derivation in Figure 7.

Finally, in Appendix D.4 we convert our first principles description of why LEARNPARAMS works above into a theoretical statement, bounding the difference between $A^t$ learned from LEARNPARAMS and $A^{t\star}$.

**1. Naive approach with full training sweeps**

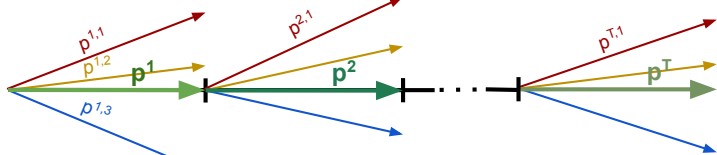

**2. Shorten training sweeps**

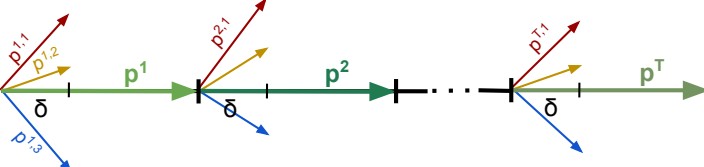

**3. Aioli: interleave training sweeps**

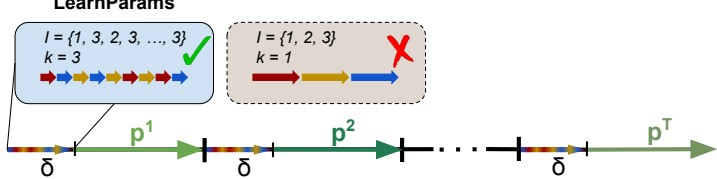

Figure 7: Derivation of AIOLI. Top: a naive high-cost approach where training sweeps are conducted to fit $A^t$ at each round (Appendix D.1). Middle: a modification that shortens the training sweeps used to learn $A^t$ (Appendix D.2). Bottom: a final modification that interleaves the sweep mixtures at a high frequency (large $k$) in one single run, enabling AIOLI's LEARNPARAMS to require no additional training (Appendix D.3).

## D.1  NAIVE TRAINING SWEEP APPROACH

This approach is depicted in Figure 7 (top). By conducting a training sweep over round $t$, we can use a linear system of equations to estimate $A^t$ from the linear dynamic mixing law $L_{\text{val},i}^{t+1}(\boldsymbol{p}) = L_{\text{val},i}^t(\boldsymbol{p}) - \sum_{j=1}^m A_{ij}^t p_j^t$. Let $p^{t,1}, p^{t,2}, ... p^{t,m} \in \triangle^m$ comprise a training sweep over the duration of round $t$. First, we checkpoint the model $f^t$, and for simplicity denote $f^t$'s validation loss on group $i$ as $L_{\text{val},i}^t$. For each $p^{t,j}$, we train $f^t$ for the entire round using $p^{t,j}$. We then record how much the validation loss on each group changes, $L_{\text{val},i}^t - L_{\text{val},i}^{t+1}(p^{t,j})$ for all $i \in [m]$. By the end of this procedure on each $p^{t,j}$, we have the following system of equations for each $i \in [m]$:

$$\sum_{j=1}^m A_{ij}^t p_j^{t,1} = L_{\text{val},i}^t - L_{\text{val},i}^{t+1}(p^{t,1}) \tag{13}$$

$$\sum_{j=1}^m A_{ij}^t p_j^{t,2} = L_{\text{val},i}^t - L_{\text{val},i}^{t+1}(p^{t,2})$$

$$\vdots$$

$$\sum_{j=1}^m A_{ij}^t p_j^{t,m} = L_{\text{val},i}^t - L_{\text{val},i}^{t+1}(p^{t,m})$$

This is a system of linear equations with $m$ unknowns: $A_{i1}, ..., A_{im}$. We can write it in matrix form as:

$$\begin{bmatrix} p_1^{t,1} & p_2^{t,1} & ... & p_m^{t,1} \\ p_1^{t,2} & p_2^{t,2} & ... & p_m^{t,2} \\ \vdots & & & \\ p_1^{t,m} & p_2^{t,m} & ... & p_m^{t,m} \end{bmatrix} \begin{bmatrix} A_{i1}^t \\ A_{i2}^t \\ \vdots \\ A_{im}^t \end{bmatrix} = \begin{bmatrix} L_{\text{val},i}^t - L_{\text{val},i}^{t+1}(p^{t,1}) \\ L_{\text{val},i}^t - L_{\text{val},i}^{t+1}(p^{t,2}) \\ \vdots \\ L_{\text{val},i}^t - L_{\text{val},i}^{t+1}(p^{t,m}) \end{bmatrix} \tag{14}$$

Let $P \in \mathbb{R}^{m \times m}$ be the leftmost matrix and $\beta_i \in \mathbb{R}^m$ be the vector on the right hand side. Then, we can write $A_i^t = P^{-1}\beta_i$. We solve this system for each $i \in [m]$ to obtain $A^t$.

The advantage of this method is that it directly estimates the optimal $A^{t\star}$ that is used in the mixing law. However, it requires $m$ sweeps per round, because the key quantity we must observe to learn $A^t$ is $L_{\text{val},i}^t - L_{\text{val},i}^{t+1}(\boldsymbol{p})$: the change in loss after training through the *entire* round $t$. As a result, this approach requires $m$ extra full training runs to learn $A^t$. Below, we will describe how we can compute cheaper alternatives to $L_{\text{val},i}^t - L_{\text{val},i}^{t+1}(\boldsymbol{p})$.

## D.2 Modification 1: shortening training sweeps

This modification is depicted in Figure 7 (middle). A simple way to reduce the number of extra training runs needed to estimate $A^t$ is to train on each mixture $p^{t,j}$ for less than a round. Let $\delta$ denote the fraction of the round we use for the training sweep. Then, our system of equations in 14 uses $L_{\text{val},i}^t - L_{\text{val},i}^{t+\delta}(p^{t,j})$; we simply record the loss difference over $\delta$ of the round rather than the entire round, and use this to solve for $A^t$. Now, this approach effectively requires $m\delta$ extra training runs; however, this cost is still linear in the number of data groups. Moreover, there is some inaccuracy incurred by using $\delta$ of a round to approximate the entire round.

## D.3 Modification 2: "interleaving" training sweeps

This modification is depicted in Figure 7 (bottom). Our final modification to derive LEARNPARAMS is to convert the training sweep—where we checkpoint the model and execute $m$ separate runs for $\delta$ of a round—into one round without requiring any checkpointing or rolling back of training. Our intuition is that if we interleave different mixtures sequentially at a high frequency, we can simulate executing these mixtures simultaneously. This is similar to a concept in signal processing called time-division multiplexing, in which two or more signals or bit streams are transferred appearing simultaneously as sub-channels in one communication channel, but are physically taking turns on the channel[1].

Formally, we break down the $\delta S/T$ steps allocated for learning $A^t$ into $K$ intervals, where $K = mk$ and $k$ is the number of sweeps per mixture. We construct an interleaved order of $p^{t,1},...,p^{t,m}$ over these $K$ intervals, and we denote their index order as $\mathcal{I} \in [m]^K$. Let $\mathcal{I}_\tau$ denote the mixture at the $\tau$th position in $\mathcal{I}$. We can denote the model at the end of each interval as $t + \delta/K, t + 2\delta/K, ..., t + \delta$. During the $\tau$th interval, we train on one $p^{t,\mathcal{I}_\tau}$ and observe the change in loss, $L_{\text{val},i}(f^{t+(\tau-1)\delta/K}) - L_{\text{val},i}(f^{t+\tau\delta/K})$ for each validation group $i$. Let $\mathcal{T}_j = \{\tau : \mathcal{I}_\tau = j\}$ be all the intervals where $p^{t,j}$ is assigned. We approximate $L_{\text{val},i}^t - L_{\text{val},i}^{t+1}(p^{t,j})$ with $\frac{1}{|\mathcal{T}_j|}\sum_{\tau \in \mathcal{T}_j}^k L_{\text{val},i}(f^{t+(\tau-1)\delta/K}) - L_{\text{val},i}(f^{t+\tau\delta/K})$. These approximated loss differences are then used to recover $A^t$ from the system of linear equations.

Lastly, note that the choice of $k$ controls the interleaving frequency and the bias of the estimated $A^t$. Suppose that $k = 1$. This means that each mixture is only assigned to one interval, and this could be at the beginning, middle, or end of the $\delta$ round. Then, the change in loss is a poor approximation of the original quantity $L_{\text{val},i}^t - L_{\text{val},i}^{t+1}(\boldsymbol{p})$ due to dependence on time. However, as we increase $k$, the mixture $p^{t,j}$ will be trained on in the beginning, middle, and end of the $\delta$ round, allowing for a less time-biased estimate of the loss change.

With this modification, LEARNPARAMS now requires *no extra training*. However, there are still some performance tradeoffs. First, in order to save compute, our estimate of $A^t$ via the shortened interleaved sweeps is less accurate than the naive approach. Second, without rolling back training, AIOLI has both an "explore" and "exploit" phase, where the former learns $A^t$ over $\delta$ of the round and the latter uses $A^t$ to set $p^t$ and mix data accordingly for the remainder of the round. If $\delta$ is large, the estimate of $A^t$ may be relatively more accurate. However, training for longer on the sweep mixtures $p^{t,1},...p^{t,m}$ may be suboptimal for the performance of the model. Moreover, the training duration that utilizes the $p^t$ that is updated using the more accurate $A^t$ is now shortened. Therefore, adjusting $\delta$ is key to ensuring that $A^t$ is accurate and the model performs well.

---

[1] https://en.m.wikipedia.org/wiki/Time-division_multiplexing

### D.4 THEORETICAL RESULTS

We now provide a bound on the difference between $A^{t\star}$, the optimal mixing law parameter, and $A^{t\text{Aioli}}$, the parameters that LEARNPARAMS estimates.

Our proof approach follows a similar break down to sections D.1, D.2, and D.3. Let $\hat{A}^t$ be the parameters learned from the naive approach in Appendix D.1, where we fit the parameter from full training sweeps over the round. Note that the difference between $\hat{A}^t$ and $A^{t\star}$ in this case is due to finite-sample regression (i.e., over the $m$ runs rather than the entire population of mixture proportions). Let $\hat{A}^{t\delta}$ be the parameters learned from the shortened approach in Appendix D.2, where we fit the parameter from shortened training sweeps. The difference between $\hat{A}^t$ and $\hat{A}^{t\delta}$ comes from variation in how the losses when trained on each mixture evolve from $t+\delta$ to $t+1$. Finally, let $\hat{A}^{t\text{Aioli}}$ be the parameters learned from LEARNPARAMS, which interleaves different proportions throughout $\delta$. The difference between $\hat{A}^{t\delta}$ and $\hat{A}^{t\text{Aioli}}$ comes from how frequent the interleaving of mixtures is.

Our goal is to find an upper bound on $\|A^{t\star}-\mu\hat{A}^{t\text{Aioli}}\|_2$, since differences in magnitudes can be offset by adjusting $\eta$. Using the triangle inequality, we now have the following decomposition:

$$\|A^{t\star}-\mu\hat{A}^{t\text{Aioli}}\|_2 \leq \underbrace{\|A^{t\star}-\hat{A}^t\|_2}_{\text{OLS error}}+\underbrace{\|\hat{A}^t-\nu\hat{A}^{t\delta}\|_2}_{\text{Error from }\delta}+\underbrace{\|\nu\hat{A}^{t\delta}-\mu\hat{A}^{t\text{Aioli}}\|_2}_{\text{Error from interleaving}} \tag{15}$$

Below, we state a set of conditions we use to bound all three quantities. Then, we present our main result, Theorem 2, before providing our proof.

**Setup for OLS error.** We frame the error from $\hat{A}^t$ being learned on finite samples as the "random design" prediction error of ordinary least squares regression (Hsu et al., 2012). We specify the data model (covariates and response variable) and use it to formally construct $A^{t\star}$ and $\hat{A}^t$.

Let the covariate $p \in \triangle^m$ be a random vector. We have $m$ response variables, $L_{\text{val},i}^t - L_{\text{val},i}^{t+1}(p)$, and $m$ separate regression problems. Note here that we have dropped the remainder of the dynamic $\boldsymbol{p}$ in the loss notation in order to model that only the mixture at time $t$, $p$, is the covariate. For simplicity, we consider a single regression problem over the $i$th validation group.

We are given $n$ samples, $(p^{t,1},L_{\text{val},i}^t - L_{\text{val},i}^{t+1}(p^{t,1})),...,(p^{t,n},L_{\text{val},i}^t - L_{\text{val},i}^{t+1}(p^{t,n}))$ to estimate each $A_i^\star$ (in our algorithm, we set $n=m$). We make the following assumptions that define $A^{t\star}$ and our data-generating model.

**Condition 1** (Ordinary least squares data model). *Suppose the following hold true regarding the true data model of $p$ and $L_{val,i}^t - L_{val,i}^{t+1}(p)$.*

1. *$p$ is drawn from a centered Dirichlet distribution. That is, $p \sim Dirichlet(\alpha_1,...,\alpha_m)$ where $\alpha_j = \alpha \geq 1$ for all $j \in [m]$.*

2. *The true loss-proportion relationship follows the form,*

$$L_{val,i}^t - L_{val,i}^{t+1}(p) = \sum_{j=1}^m A_{ij}^{t\star}p + \varepsilon_{OLS}, \tag{16}$$

*for some $A_i^{t\star} \in \mathbb{R}^m$, where $\varepsilon_{OLS}$ is a zero-mean noise term. That is, we assume no approximation error in our data-generating model.*

3. *The noise term $\varepsilon_{OLS}$ is Subgaussian, meaning that there exists a $\sigma_{OLS} \geq 0$ such that almost surely,*

$$\mathbb{E}[\exp(\eta\varepsilon_{OLS})] \leq \exp(\eta^2\sigma_{OLS}^2/2) \,\forall \eta \in \mathbb{R}. \tag{17}$$

We assume that $p$ is drawn from the Dirichlet distribution with $\alpha \geq 1$, since exponential gradient descent uses an entropy regularization term that pushes proportions away from the edges of the simplex. We assume that there is no approximation error in the linear mixing law, since we have established that it is of high fidelity.

Now, we formally define $\hat{A}^t$. Let $\Sigma$ denote the covariance $\mathbb{E}\left[pp^\top\right]$. Let $\hat{\Sigma}$ be the sample-level estimate of the covariance, and let $\hat{\mathbb{E}}[]$ express sample-level expectation on our size $n$ dataset. Then, our ordinary least squares estimator is

$$\hat{A}_i^t = \hat{\Sigma}^{-1}\hat{\mathbb{E}}\left[p(L_{\text{val},i}^t - L_{\text{val},i}^{t+1}(p))\right] \tag{18}$$

$\hat{A}_i$ is an unbiased estimate of $A_i^\star$. Moreover, for $n = m$, this estimator is equivalent to setting $\hat{A}_i = P^{-1}[L_{\text{val},i}^t - L_{\text{val},i}^{t+1}(p^{t,1}),...,L_{\text{val},i}^t - L_{\text{val},i}^{t+1}(p^{t,n})]^\top$ over the linear system of $m$ equations, where $P$ is defined in LEARNPARAMS.

**Setup for error from $\delta$.**     We can express $\hat{A}^{t\delta}$ as the solution to a linear system of equations where the response variable is computed over shortened runs. First, we abuse notation from LEARNPARAMS and define $\beta_i \in \mathbb{R}^n$ to be the response variable vector over our $n$ samples, where $\beta_{ij} = L_{\text{val},i}^t - L_{\text{val},i}^{t+1}(p^{t,j})$. Then, we recall that $\hat{A}_i^t = P^{-1}\beta_i$, where $P$ satisfies $P_{ij} = \frac{\varepsilon}{m}$ for $i \neq j$ and $P_{ii} = 1 - \varepsilon + \frac{\varepsilon}{m}$.

Next, define $\beta_i^\delta \in \mathbb{R}^n$, where $\beta_{ij}^\delta = L_{\text{val},i}^t - L_{\text{val},i}^{t+\delta}(p^{t,j})$—the loss difference in group $i$ when training on $p^{t,j}$ over $\delta$ of a round. We have that

$$\hat{A}^{t\delta} = P^{-1}\beta_i^\delta, \tag{19}$$

$$\beta_{ij}^\delta = L_{\text{val},i}^t - L_{\text{val},i}^{t+\delta}(p^{t,j}) \,\forall j \in [n] \tag{20}$$

The next condition characterizes the relationship between $\beta_i$ and $\beta_\delta$, connecting the full round's loss difference to the shortened loss difference.

**Condition 2** (Linear + noise time model)**.** *There exists $c_\delta \in (0,1)$ and $\varepsilon_\delta \sim \mathcal{N}(0,(1-\delta)^2\sigma_\delta^2)$ such that*

$$\beta_i^\delta = c_\delta\beta_i + \varepsilon_\delta \,\forall i \in [m]. \tag{21}$$

Intuitively, this means that when shortening a training run from one round to $\delta$ of a round, the loss change is scaled by a linear factor $c_\delta$ and then adjusted by some Gaussian noise. $\varepsilon_\delta$ captures how much variance there is in the shortened training runs relative to the full training runs. Note that as $\delta$ approaches 1, $c_\delta$ approaches 1 and $\varepsilon_\delta$ decreases in variance.

**Setup for error from interleaving.**     We define $\hat{A}^{t\text{Aioli}}$ now. Recall that we have taken $\delta$ of the round and split it into $K$ equal intervals. During each interval, we train on some mixture $p^{t,j}$. Let the order of interleaved indices be captured by $\mathcal{I} \in [m]^K$. $\mathcal{I}_\tau$ is the index of the mixture that is trained on in the $\tau$th interval, and $\mathcal{T}_j = \{\tau : \mathcal{I} = j\}$ are all the intervals for which the model is trained using $p^{t,j}$. $\mathcal{I}$ is constructed such that $|\mathcal{T}_j| = k$ for all $j \in [n]$. Moreover, we assume that each set of indices for $j$ is selected with equal probability (i.e., $\Pr(\mathcal{T}_j) = 1/\binom{K}{k}$).

Due to the dynamic nature of this training procedure, we abuse loss notation to capture what the model has already been trained on so far. Let $L_{\text{val},i}^{t+\tau\delta/K}(p|[\cdot])$ denote that the model up to interval $\tau$ was trained on $[\cdot]$. Let $\beta_i^{\mathcal{I}} \in \mathbb{R}^n$ be the response variable vector over $n$ samples, where each $\beta_{ij}$ is equal to $\frac{1}{k}\sum_{\tau \in \mathcal{T}_j}\left(L_{\text{val},i}^{t+(\tau-1)\delta/K}(p^{t,\mathcal{I}_{\tau-1}}|\mathcal{I}_{1:\tau-2}) - L_{\text{val},i}^{t+\tau\delta/K}(p^{t,j}|\mathcal{I}_{1:\tau-1})\right)$. Here, we have annotated the losses to show that the loss differences are always computed before and after training an interval using $p^{t,j}$, even though the remaining intervals are trained on other mixtures. Using this definition of $\beta_i^{\mathcal{I}}$, we have

$$\hat{A}^{t\text{Aioli}} = P^{-1}\beta_i^{\mathcal{I}}, \tag{22}$$

$$\beta_{ij}^{\mathcal{I}} = \frac{1}{k}\sum_{\tau \in \mathcal{T}_j}\left(L_{\text{val},i}^{t+(\tau-1)\delta/K}(p^{t,\mathcal{I}_{\tau-1}}|\mathcal{I}_{1:\tau-2}) - L_{\text{val},i}^{t+\tau\delta/K}(p^{t,j}|\mathcal{I}_{1:\tau-1})\right). \tag{23}$$

Moreover, we can write $\beta_{ij}^{\delta}$ in terms of interval-level loss differences:

$$\beta_{ij}^{\delta} = L_{\text{val},i}^{t} - L_{\text{val},i}^{t+\delta}(p^{t,j}) = \left(L_{\text{val},i}^{t} - L_{\text{val},i}^{t+\delta/K}(p^{t,j})\right) + \left(L_{\text{val},i}^{t+\delta/K}(p^{t,j}) - L_{\text{val},i}^{t+2\delta/K}(p^{t,j}|p^{t,j})\right) \quad (24)$$

$$+ \cdots + \left(L_{\text{val},i}^{t+(K-1)\delta/K}(p^{t,j}|p^{t,j}) - L_{\text{val},i}^{t+\delta}(p^{t,j}|p^{t,j})\right). \quad (25)$$

We make the following assumption relating $\beta_{ij}^{\delta}$ and $\beta_{ij}^{\mathcal{I}}$.

**Condition 3** (Independent loss differences)**.** *Fix a* $j \in [n]$. *For any interval* $\tau \in \mathcal{T}_j$,

$$L_{val,i}^{t+(\tau-1)\delta/K}(p^{t,\mathcal{I}_{\tau-1}}|\mathcal{I}_{1:\tau-2}) - L_{val,i}^{t+\tau\delta/K}(p^{t,j}|\mathcal{I}_{1:\tau-1}) = L_{val,i}^{t+(\tau-1)\delta/K}(p^{t,j}|p^{t,j}) - L_{val,i}^{t+\tau\delta/K}(p^{t,j}|p^{t,j})$$

Intuitively, this means that the loss difference before and after we train on $p^{t,j}$ at interval $\tau$ is *independent* of what we trained on before this point; this quantity only depends on $p^{t,j}$ and $\tau$.

**Main result.** We are now ready to present our main theorem that bounds $\|A^{t\star} - \mu\hat{A}^{t\text{Aioli}}\|_2$. We provide a few extra definitions, as well as reminders of key terms.

Recall that $m$ is the number of domains and $n$ is the number of samples used to fit the training sweep (we use $n=m$ in practice). $\alpha$ is the parameter of the Dirichlet distribution for generating mixtures. $\sigma_{\text{OLS}}^2$ is the Subgaussian parameter for the noise in the mixing law. $\delta$ is the proportion of the round allocated for learning $A^t$. $\sigma_{\delta}^2$ and $c_{\delta}$ are the noise and slope in the linear model, respectively, for how loss change over $\delta$ of a round is expressed in terms of loss change over the entire round. $\varepsilon$ is the one-hot vector smoothing factor used in LEARNPARAMS. $K$ is the total number of intervals in $\delta$, and $k$ is the number of intervals per mixture, where $kn=K$. Let $a_{ij} = \max_{\tau} L_{\text{val},i}^{t+\tau\delta/K}(p^{t,j}|p^{t,j})$ and let $b_{ij} = \min_{\tau} L_{\text{val},i}^{t+\tau\delta/K}(p^{t,j}|p^{t,j})$. Define $r = \max_{i,j} a_{ij} - b_{ij}$.

**Theorem 2.** *Suppose that Conditions 1, 2, and 3 hold. Let $t$ satisfy $t > \max\{0, 2.6 - \log m\}$ and assume $n \geq \frac{6(\log m + t)}{m(m\alpha+1)}$. There exists some $\mu > 0$ such that the estimation error of* LEARNPARAMS *can be bounded as follows:*

$$\|A^{t\star} - \mu\hat{A}^{t\text{Aioli}}\|_2 \leq Err_{OLS} + Err_{\delta} + Err_{\mathcal{I}} \ \ w.p. \ \geq 1 - 3m\exp(-t) - \exp\left(-\frac{mt^2(1-t)}{4}\right) - nmt, \quad (26)$$

*where there exists some finite $c > 0$ and*

$$Err_{OLS} = \frac{c\sigma_{OLS}^2 m(m+2\sqrt{mt}+2t)\sqrt{m\alpha+1}}{n} \quad (27)$$

$$Err_{\delta} = \frac{(1-\delta)^2\sigma_{\delta}^2 m^{3/2}(1+t)}{(1-\varepsilon)c_{\delta}^2} \quad (28)$$

$$Err_{\mathcal{I}} = \frac{rK}{c_{\delta}(1-\varepsilon)}\sqrt{\frac{nm\ln(1/t)\left(1-\frac{k-1}{K}\right)}{2k}} \quad (29)$$

We briefly discuss each of the three terms:

- $Err_{\text{OLS}}$: this term describes the error incurred when learning $A^t$ over finite number of samples. We see that there are several quantities controlling this error. As the number of samples $n$ increases, this error goes to 0. Similarly, if the linear dynamic mixing law is not very noisy, then $\sigma_{\text{OLS}}^2$ is small and the error will be negligible. Lastly, note the dependence on $\alpha$, the key parameter dictating how sharp the Dirichlet distribution is.

- $Err_{\delta}$: in this term, we can see that if $\delta$ is close to 1, the full round, then the error is negligible. Moreover, if the shortened rounds are not noisy and $\sigma_{\delta}^2$ is low, then the error is also small. Lastly, observe the $1-\varepsilon$ in the denominator. Intuitively, this means that if the $p^{t,1}, ... p^{t,m}$ are spread out, things are easier to learn.

- $\text{Err}_{\mathcal{I}}$: this term is controlled by the relationship between $k$, how many intervals you use out of $K$ to train on each $p^{t,j}$, and $K$, the total number of intervals. If $k \approx K$, then error is roughly constant. However, if $k$ is small, the error from interleaving can be high because we are only using a fraction of $\delta$ for estimating loss differences.

### D.5 PROOF OF THEOREM 2

We break down our proof into three lemmas that bound $\|A^{t\star} - \hat{A}^t\|_2$, $\|\hat{A}^t - \nu \hat{A}^{t\delta}\|_2$, and $\|\nu \hat{A}^{t\delta} - \mu \hat{A}^{t\text{Aioli}}\|_2$.

#### D.5.1 OLS ERROR BOUND

First, we restate Theorem 11 from Hsu et al. (2012), which we apply to obtain our bound on $\|A^{t\star} - \hat{A}^t\|_2$.

**Theorem 3** (Hsu et al. (2012)). *Denote $\|x\|_{\Sigma}$ to be $\sqrt{x^\top \Sigma x}$. Assume that Condition 1 holds. Let $t$ satisfy $t > \max\{0, 2.6 - \log m\}$. Let $\rho_0$ satisfy $\frac{\|\Sigma^{-1/2}p\|_2}{\sqrt{m}} \le \rho_0$. Assume that the number of samples $n$ satisfies $n \ge 6\rho_0^2 m(\log m + t)$. Then, with probability at least $1 - 3e^{-t}$, we have*

$$\|A_i^{t\star} - \hat{A}_i^t\|_{\Sigma} \le \frac{2}{1 - \delta_s} \cdot \frac{\sigma_{OLS}^2(m + 2\sqrt{mt} + 2t)}{n}, \tag{30}$$

*where $\delta_s = \sqrt{\frac{4\rho_0^2 m(\log m + t)}{n}} + \frac{2\rho_0^2 m(\log m + t)}{3n}$.*

To adapt this theorem to our setting, we must compute $\rho_0$ and $\delta_s$, and convert the bound into a result on the 2-norm of the matrix difference.

**Lemma 4** (OLS random design bound). *Let $t$ satisfy $t > \max\{0, 2.6 - \log m\}$. Assume Condition 1 holds, and the number of samples $n$ satisfies $n \ge \frac{6(\log m + t)}{m(m\alpha + 1)}$. Then, with probability at least $1 - 3me^{-t}$, we have*

$$\|\hat{A}^t - A^{t\star}\|_2 \le \frac{2}{1 - \delta_s} \cdot \frac{\sigma_{OLS}^2 m\sqrt{m}(m + 2\sqrt{mt} + 2t)(m\alpha + 1)}{n}, \tag{31}$$

*where $\delta_s = 2\left(\sqrt{\frac{\log m + t}{nm(m\alpha + 1)}} + \frac{\log m + t}{3nm(m\alpha + 1)}\right)$.*

*Proof.* We first compute $\rho_0$ and $\delta_s$. From the triangle inequality and definition of the norm, we have that $\frac{\|\Sigma^{1/2}p\|_2}{\sqrt{m}} \le \frac{\|\Sigma\|_2^{1/2}\|p\|_2}{\sqrt{m}}$. Using Lemma 7 and the fact that $\|p\|_2 \le 1$, we can set $\rho_0 = \frac{\sqrt{\frac{1}{m(m\alpha + 1)}}}{\sqrt{m}} = \frac{1}{m\sqrt{(m\alpha + 1)}}$. Then, we have that

$$\delta_s = 2\left(\sqrt{\frac{\log m + t}{nm(m\alpha + 1)}} + \frac{\log m + t}{3nm(m\alpha + 1)}\right). \tag{32}$$

To convert the bound in (30) into a bound on $\|\hat{A}_i^t - A_i^{t\star}\|_2$, we note that $\|\hat{A}_i^t - A_i^{t\star}\|_2 = \|(\hat{A}_i^t - A_i^{t\star})\Sigma^{1/2}\Sigma^{-1/2}\|_2 \le \|(\hat{A}_i^t - A_i^{t\star})\Sigma^{1/2}\|_2\|\Sigma^{-1/2}\|_2 \le \|\hat{A}_i^t - A_i^{t\star}\|_{\Sigma}\|\Sigma^{-1/2}\|_2$. Then, using Lemma 7, we have

$$\|\hat{A}_i - A_i^{\star}\|_2 \le \frac{2}{1 - \delta_s} \cdot \frac{\sigma^2(m + 2\sqrt{mt} + 2t)\sqrt{m(m\alpha + 1)}}{n}. \tag{33}$$

Then, we use $\|\hat{A}^t - A^{t\star}\|_2 \le \|\hat{A}^t - A^{t\star}\|_F = \sqrt{\sum_{i=1}^m \|\hat{A}_i^t - A_i^{t\star}\|_2^2} \le \sqrt{m}\|\hat{A}_i^t - A_i^{t\star}\|_2$. Applying a union bound, we have

$$\|\hat{A}^t - A^{t\star}\|_2 \le \frac{2}{1 - \delta_s} \cdot \frac{\sigma_{OLS}^2 m(m + 2\sqrt{mt} + 2t)\sqrt{m\alpha + 1}}{n} \quad \text{w.p. } 1 - 3me^{-t} \tag{34}$$

$\square$

### D.5.2 ERROR FROM $\delta$ BOUND

**Lemma 5.** *Define $t \in [0,1]$ and assume that Condition 2 holds. There exists a $\nu \geq 0$ such that*

$$\|\hat{A}^t - \nu \hat{A}^{t\delta}\|_2 \leq \|P^{-1}\|_2 \sqrt{\sum_{i=1}^{m} \|\varepsilon_\delta/c_\delta\|_2^2} = \|P^{-1}\|_2 \cdot \left\|\frac{\varepsilon_\delta}{c_\delta}\right\|_2^2 \cdot \sqrt{m} \tag{35}$$

*Proof.* We first simplify the left hand side and plug in the construction from Condition 2:

$$\|\hat{A}^t - \nu \hat{A}^{t\delta}\|_2 \leq \sqrt{\sum_{i=1}^{m} \|\hat{A}_i - \nu \hat{A}_i^\delta\|_2^2} \leq \|P^{-1}\|_2 \sqrt{\sum_{i=1}^{m} \|\beta_i - \nu\beta_i^\delta\|_2^2} = \|P^{-1}\|_2 \sqrt{\sum_{i=1}^{m} \|\beta_i - \nu(c_\delta\beta_i + \varepsilon_\delta)\|_2^2}. \tag{36}$$

Setting $\nu$ equal to $\frac{1}{c_\delta}$, we have

$$\|\hat{A}^t - \nu \hat{A}^{t\delta}\|_2 \leq \|P^{-1}\|_2 \sqrt{\sum_{i=1}^{m} \|\varepsilon_\delta/c_\delta\|_2^2} = \|P^{-1}\|_2 \cdot \left\|\frac{\varepsilon_\delta}{c_\delta}\right\|_2^2 \cdot \sqrt{m} \tag{37}$$

From Lemma 8, we have that $\|P^{-1}\|_2$ under our construction in LEARNPARAMS is $\frac{1}{1-\varepsilon}$. Moreover, note that $\|\varepsilon_\delta\|^2$ is equal to $(1-\delta)^2\sigma_\delta^2\chi_m^2$, where $\chi_m^2$ is the Chi-squared distribution with $m$ degrees of freedom. Using a Chernoff bound, we have that $\Pr(\|\varepsilon_\delta\|^2 \leq (1-\delta)^2\sigma_\delta^2 m(1+t)) \leq 1 - \exp(-\frac{m}{2}(t + \ln\frac{1}{1+t}))$ for some $t > 0$. Putting everything together, we have

$$\|\hat{A} - \mu_1\hat{A}^\delta\|_2 \leq \frac{(1-\delta)^2\sigma_\delta^2 m\sqrt{m}(1+t)}{(1-\varepsilon)c_\delta^2} \text{ w.p. } 1 - \exp\left(-\frac{m}{2}\left(t + \ln\frac{1}{1+t}\right)\right) \tag{38}$$

$\square$

### D.5.3 ERROR FROM INTERLEAVING BOUND

**Lemma 6.** *Assume that Condition 3 holds. Let $a_{ij} = \max_\tau L_{val,i}^{t+\tau\delta/K}(p^{t,j}|p^{t,j})$ and let $b_{ij} = \min_\tau L_{val,i}^{t+\tau\delta/K}(p^{t,j}|p^{t,j})$. Define $r = \max_{i,j} a_{ij} - b_{ij}$. Then,*

$$\|\nu\hat{A}^{t\delta} - \mu\hat{A}^{tAioli}\| \leq \frac{rK}{c_\delta(1-\varepsilon)}\sqrt{\frac{nm\ln(1/t)\cdot(1-\frac{k-1}{K})}{2k}} \text{ w.p. } \geq 1 - nmt. \tag{39}$$

*Proof.* We can write $\|\nu\hat{A}^{t\delta} - \mu\hat{A}^{tAioli}\|$ as

$$\|\nu\hat{A}^{t\delta} - \mu\hat{A}^{tAioli}\| \leq \nu\|P^{-1}\|_2\sqrt{\sum_{i=1}^{m}\|\beta_i^\delta - \frac{\mu}{\nu}\beta_i^\mathcal{I}\|_2^2} = \frac{\nu}{1-\varepsilon}\sqrt{\sum_{i=1}^{m}\|\beta_i^\delta - \frac{\mu}{\nu}\beta_i^\mathcal{I}\|_2^2}. \tag{40}$$

We write $\beta_{ij}^\delta = x_{1|j} + x_{2|j} + \cdots + x_{K|j}$, where $x_{\tau|j} = L_{val,i}^{t+(\tau-1)\delta/K}(p^{t,j}|p^{t,j}) - L_{val,i}^{t+\tau\delta/K}(p^{t,j}|p^{t,j})$. Similarly, we write $\beta_{ij}^\mathcal{I} = x_{1|\mathcal{I}} + x_{2|\mathcal{I}} + \cdots + x_{K|\mathcal{I}}$, where $x_{\tau|\mathcal{I}} = L_{val,i}^{t+(\tau-1)\delta/K}(p^{t,j}|\mathcal{I}_{1:\tau-2}) - L_{val,i}^{t+\tau\delta/K}(p^{t,j}|\mathcal{I}_{1:\tau-1})$. From Condition 3, we have that $x_{\tau|j} = x_{\tau|\mathcal{I}}$ for any $\tau \in \mathcal{T}_j$. This means that

$$\beta_{ij}^\mathcal{I} = \frac{1}{k}\sum_{\tau \in \mathcal{T}_j} x_{\tau|j}. \tag{41}$$

This gives us an elegant way of comparing $\beta_{ij}^{\mathcal{I}}$ and $\beta_{ij}^{\delta}$: in expectation, we have that $\mathbb{E}\left[\beta_{ij}^{\mathcal{I}}\right] = \frac{\beta_{ij}^{\delta}}{K}$. Then, $\|\beta_i^{\delta} - \frac{\mu}{\nu}\beta_i^{\mathcal{I}}\|^2 = \sum_{j=1}^n (\beta_{ij}^{\delta} - \frac{\mu}{\nu}\beta_{ij}^{\mathcal{I}})^2 = \sum_{j=1}^n (\sum_{\tau=1}^K x_{\tau|j} - \frac{\mu}{\nu}\frac{1}{k}\sum_{\tau\in\mathcal{T}_j} x_{\tau|j})^2$. If we set $\mu/\nu$ to be equal to $K$, then we get

$$\|\beta_i^{\delta} - \frac{\mu}{\nu}\beta_i^{\mathcal{I}}\|^2 = K^2 \sum_{j=1}^n \Big( \frac{1}{K}\sum_{\tau=1}^K x_{\tau|j} - \frac{1}{k}\sum_{\tau\in\mathcal{T}_j} x_{\tau|j} \Big)^2 \tag{42}$$

Applying Theorem 5 and a union bound, this becomes

$$\|\beta_i^{\delta} - \frac{\mu}{\nu}\beta_i^{\mathcal{I}}\|^2 \leq \frac{nK^2\ln(1/t)\cdot(1-\frac{k-1}{K})r^2}{2k} \ w.p. \geq 1-nt \tag{43}$$

where $r$ is the range of $x_{\tau|j}$. Finally, we plug this into the original bound and use $\nu = \frac{1}{c_\delta}$ and a union bound again.

$$\|\nu\hat{A}^{t\delta} - \mu\hat{A}^{t\text{Aioli}}\| \leq \frac{rK}{c_\delta(1-\varepsilon)}\sqrt{\frac{nm\ln(1/t)\cdot(1-\frac{k-1}{K})}{2k}} \ w.p. \geq 1-nmt. \tag{44}$$

$\square$

### D.5.4 AUXILIARY LEMMAS

**Theorem 4** (Horn & Johnson (1985)). *If matrix $A \in \mathbb{R}^{m\times m}$ has eigenvalues $\lambda, \lambda_2, ..., \lambda_m$ and $x$ is the eigenvector such that $Ax = \lambda x$, then for any $v \in \mathbb{R}^m$, $A + xv^\top$ has eigenvalues $\lambda + v^\top x, \lambda_1, ..., \lambda_m$.*

**Lemma 7.** *For $p \sim Dirichlet(\alpha_1, ..., \alpha_m)$ where $\alpha_i = \alpha$ for all $i \in [m]$, the L2 norm of the covariance matrix $\Sigma = \mathbb{E}\left[pp^\top\right]$ is*

$$\|\Sigma\|_2 = \frac{1}{m(m\alpha+1)}. \tag{45}$$

*Proof.* First, the Dirichlet covariance matrix with constant $\alpha_i$'s has the following form:

$$\Sigma_{ij} = \begin{cases} \frac{-1}{m^2(m\alpha+1)} & i \neq j \\ \frac{m-1}{m^2(m\alpha+1)} & i = j \end{cases} \tag{46}$$

We can write this covariance matrix as a sum of a rank one matrix and a diagonal matrix:

$$\Sigma = \text{diag}\Big(\frac{1}{m(m\alpha+1)}\Big) - \frac{1}{m^2(m\alpha+1)}\mathbf{1}\mathbf{1}^\top$$

Note that the diagonal matrix has all eigenvalues equal to $\frac{1}{m(m\alpha+1)}$, and every vector in $\mathbb{R}^m$ is an eigenvector.

Next, we apply Theorem 2.4.10.1 from Horn & Johnson (1985), which is restated in Theorem 4. We set $A = \text{diag}(\frac{1}{m(m\alpha+1)})$, $x = \mathbf{1}$, $v = -\frac{1}{m^2(m\alpha+1)}\mathbf{1}$. Then, the eigenvalues of $\Sigma$ are $\frac{1}{m(m\alpha+1)} + m\frac{-1}{m^2(m\alpha+1)} = 0$ (multiplicity 1) and $\frac{1}{m(m\alpha+1)}$ (multiplicity $m-1$). Therefore, we have that $\|\Sigma\|_2 = \frac{1}{m(m\alpha+1)}$. $\square$

**Lemma 8.** *The matrix $P \in \mathbb{R}^{m\times m}$ is defined in* LEARNPARAMS *as:*

$$P_{ij} = \begin{cases} 1-\varepsilon+\frac{\varepsilon}{m} & i=j \\ \frac{\varepsilon}{m} & i\neq j \end{cases}, \tag{47}$$

*where $\varepsilon$ is a one-hot correction factor. Then,*

$$\|P^{-1}\|_2 = \frac{1}{1-\varepsilon}. \tag{48}$$

*Proof.* First, observe that we can write $P$ as the sum of a rank one and diagonal matrix.

$$P = \frac{\varepsilon}{m} \cdot \mathbf{1}\mathbf{1}^\top + \mathrm{diag}(1-\varepsilon) \tag{49}$$

The diagonal matrix has all eigenvalues equal to $\frac{\varepsilon}{m}$. Next, we apply Theorem 2.4.10.1 from Horn & Johnson (1985), restated in Theorem 4, to the sum of the diagonal and rank-one matrix. The eigenvalues are $1-\varepsilon+\varepsilon=1$ (multiplicity 1) and $1-\varepsilon$ (multiplicity $n-1$). The eigenvalues of $P^{-1}$ are then $1$ and $\frac{1}{1-\varepsilon}$, and $\frac{1}{1-\varepsilon}$ is the largest eigenvalue. □

**Theorem 5** (Serfling (1974)). *Let $x_1,...,x_K$ be a finite list of values. We draw a sample of size $k$ without replacement, each with equal probability—denote this as $X_1,...,X_k$. Define $a=\min_i x_i$ and $b=\max_i x_i$. Then,*

$$\Pr\left(\frac{1}{k}\sum_{i=1}^k X_k - \frac{1}{K}\sum_{i=1}^K x_K \geq t\right) \leq \exp\left(-\frac{2kt^2}{(1-(k-1)/K)(b-a)^2}\right). \tag{50}$$

*This can also be written as*

$$\frac{1}{k}\sum_{i=1}^k X_k - \frac{1}{K}\sum_{i=1}^K x_K \leq \sqrt{\frac{\ln(1/\delta)\cdot(1-\frac{k-1}{K})(b-a)^2}{2k}} \ w.p. \geq 1-\delta \tag{51}$$

## E    EXPERIMENTAL DETAILS

### E.1    DATA

To obtain a test set, we shuffle and split the validation set from SlimPajama-6B (Soboleva et al., 2023; Yoon, 2023) in half.

To perform training sweeps and emulate grid searches in static settings for $m=3,7$, we oversampled from the Dirichlet with $\alpha=1$ by $4x$ the number of points and then hierarchically merged closest points into a centroid until we obtained $x$ points. For example, to obtain 10 points in the 7-dimensional simplex for SlimPajama-full, we would sample 40 points in the simplex and hierarchically merge closest points until 10 points remain. This is to ensure that near-duplicate $\boldsymbol{p}$'s are not included in the sweep. This procedure is used in Grid Search (GS) and DML in Section 6 and in our analysis in Section 4

#### E.1.1    TRAINING

Here, we discuss the training setups for the restricted and unrestricted settings. For the $m=2,3$ settings, we train a 160M model using Pythia-160M's configuration for $S=5000$ steps and results are averaged over 5 random seeds. For $m=7$, we train a 160M model using Pythia-160M's configuration for $S=40000$ steps results are averaged over 3 random seeds. All settings use FlashAttention (Dao et al., 2022), batch size of 8, context size of 2048, and cosine learning rate decay from a starting learning rate of 5e-5 to 1e-5 with 500 steps of learning rate warmup.

For the $m=2,3$ settings, experiments were run on a NVIDIA RTX 6000 Ada Generation GPU. For the $m=7$ setting, experiments were run on a NVIDIA A100 80 GB GPU.

**Restricted versus unrestricted.**    Both the restricted and unrestricted settings share the same length of the final training runs (5000 and 40000 steps, as above). The unrestricted setting gives all methods up to 10 training runs to initialize mixing algorithm parameters, or $10S$ steps, while the restricted setting give $0.5S$ steps. See Table 9 for training budget allocations in each setting. AIOLI and stratified sampling do not use extra training runs.

### E.2    DATA MIXING METHODS

**AIOLI-specific hyperparameters**    In the unrestricted setting, we found it sometimes helpful to use an exponential moving average with proportion $\gamma$ over $A^t$ for AIOLI. Formally, the standard $p^t$ update rule in

Table 9: Training budget allocations for restricted and unrestricted settings.

| Setting | $m$ | Method | Runs within training budget |
|---|---|---|---|
| Unrestricted | 2 | DML | 10 runs, 5000 steps |
| | | Skill-it | 2 runs, 5000 steps |
| | | DoReMi | 2 runs, 5000 steps |
| | | DoGE | 1 run, 5000 steps |
| | 3 | DML | 10 runs, 5000 steps |
| | | Skill-it | 3 runs, 5000 steps |
| | | DoReMi | 2 runs, 5000 steps |
| | | DoGE | 1 run, 5000 steps |
| | 7 | DML | 10 runs, 40000 steps |
| | | Skill-it | 7 runs, 40000 steps |
| | | DoReMi | 2 runs, 40000 steps |
| | | DoGE | 1 run, 40000 steps |
| Restricted | 2 | DML | 10 runs, 250 steps |
| | | Skill-it | 2 runs, 1250 steps |
| | | DoReMi | 2 runs, 1250 steps |
| | | DoGE | 1 run, 2500 steps |
| | 3 | DML | 10 runs, 250 steps |
| | | Skill-it | 3 runs, 833 steps |
| | | DoReMi | 2 runs, 1250 steps |
| | | DoGE | 1 run, 2500 steps |
| | 7 | DML | 10 runs, 2000 steps |
| | | Skill-it | 7 runs, 2814 steps |
| | | DoReMi | 2 runs, 10000 steps |
| | | DoGE | 1 run, 20000 steps |

Algorithm 1 can be unrolled as $p_j^{t+1} \propto p_j^0 \exp(\eta \sum_{\tau=1}^t \sum_{i=1}^m A_{ij}^\tau)$, which places equal weight on every $A_{ij}^\tau$. To incorporate the EMA, we define $A_{\text{ema}}^1 = \bar{A}^1$ and $A_{\text{ema}}^t = (1-\gamma)\bar{A}^t + \gamma A_{\text{ema}}^{t-1}$. We then use the update rule $p_j^{t+1} \propto p_j^0 \exp(\eta A_{\text{ema}}^t)$. This allows AIOLI to gradually decay the contributions of $A^t$, such that the value of $p^t$ is less dependent on earlier proportions in the training.

We summarize the hyperparameters used in AIOLI, providing their default values as well as guidelines for how to set them. Refer to Algorithm 1 and 2 to see how they are used:

- Number of rounds $T$: we set this to 20 in all experiments. Larger $T$ means more frequent updates to the mixture proportions.

- Sweeps $k$: we set this to be 4 for $m=2,3$ and 2 for the full SlimPajama experiments. We did not adjust this hyperparameter otherwise. Intuitively, a larger $k$ will give a more accurate $A^t$, because this means that each $p^{i,t}$ will be trained on more frequently throughout the $\delta$ proportion of the round; however, this will also result in less of the round being allocated to exploiting $A^t$ via using $p^t$.

- $\varepsilon$ one-hot smoothing factor: we set this to be $0.75$ in all experiments. In general, $\varepsilon$ must be set between 0 and 1, where 0 results in the training sweep using one-hot mixture proportions to learn $A^t$, which means that each batch only consists of one data group and can result in poor learning dynamics. $\varepsilon = 1$, on the other hand, means that our training sweep would only consist of uniform proportions.

- EGD step size $\eta$: we sweep $\{0.1, 0.2, 0.3, 0.5\}$, with higher $\eta$ resulting in greater magnitude of the proportion update.

- Proportion of round $\delta$ dedicated to learning $A^t$: We use $\delta = 0.128, 0.288, 0.007$ for $m = 2, 3, 7$, respectively. Intuitively, a larger $\delta$ will give more accurate $A^t$ because the parameter is learned on more data, but this will also result in less of the round being allocated to exploiting $A^t$ via using $p^t$.

- EMA parameter $\gamma$: we sweep None, 0.1, 0.5. Intuitively, None means that the $p^t$ update is equally dependent on all previous $p^t$'s, while a small $\gamma = 0$ means that the $p^t$ update is only a function of the current $A^t$.

For the last three hyperparameters, $\eta, \delta, \gamma$, we used different values of them in different experiments. Tables 10, 11, 12, 13, 14, and 15 list exact values for the unrestricted and restricted settings for $m=2,3,7$. In addition, Appendix F.3 provides results on hyperparameter sensitivity for $\eta, \delta$, and $\gamma$.

Table 10: **Unrestricted** hyperparameter values for each data mixing algorithm for experiments where $m=2$ (corresponding to Table 2 results).

| Data groups | Hyperparameter | Value |
|---|---|---|
| **arXiv/SE** | · proportion of round $\delta$ | 0.128 |
| | · EGD learning rate $\eta$ | 0.2 |
| | · EMA parameter $\gamma$ | 0.1 |
| **GitHub/C4** | · proportion of round $\delta$ | 0.128 |
| | · EGD learning rate $\eta$ | 0.3 |
| | · EMA parameter $\gamma$ | 0.5 |
| **Books/SE** | · proportion of round $\delta$ | 0.128 |
| | · EGD learning rate $\eta$ | 0.1 |
| | · EMA parameter $\gamma$ | None |

Table 11: **Restricted** hyperparameter values for each data mixing algorithm for experiments where $m=2$ (corresponding to Table 3 results).

| Data groups | Hyperparameter | Value |
|---|---|---|
| **arXiv/SE** | · proportion of round $\delta$ | 0.128 |
| | · EGD learning rate $\eta$ | 0.2 |
| | · EMA parameter $\gamma$ | None |
| **GitHub/C4** | · proportion of round $\delta$ | 0.128 |
| | · EGD learning rate $\eta$ | 0.2 |
| | · EMA parameter $\gamma$ | None |
| **Books/SE** | · proportion of round $\delta$ | 0.128 |
| | · EGD learning rate $\eta$ | 0.2 |
| | · EMA parameter $\gamma$ | None |

Table 12: **Unrestricted** hyperparameter values for each data mixing algorithm for experiments where $m=3$ (corresponding to Table 2 results).

| Data groups | Hyperparameter | Value |
|---|---|---|
| **arXiv/Books/SE** | · proportion of round $\delta$ | 0.288 |
| | · EGD learning rate $\eta$ | 0.5 |
| | · EMA parameter $\gamma$ | None |
| **CommonCrawl/GitHub/Wiki** | · proportion of round $\delta$ | 0.288 |
| | · EGD learning rate $\eta$ | 0.3 |
| | · EMA parameter $\gamma$ | 0.5 |

**Baseline hyperparameters.** We consulted the original papers and implementations to determine how to set the hyperparameters for each baseline, ensuring that the updated proportions were changing significantly but not oscillating under these configurations.

- **Skill-It**: the hyperparameters are the number of rounds $T$, the EGD learning rate $\eta$, and the multiplicative weights window $w$. Our default configuration was $T=10$, $\eta=0.2$, and $w=3$. However, we made two exceptions in the unrestricted setting after conducting a sweep over $T \in \{5,10\}$ and $\eta \in \{0.1,0.2,0.5,0.8\}$; for GitHub/C4, we used $T=5$ and $\eta=0.1$, and for Books/StackExchange, we used $\eta=0.8$.
- **DoReMi**: the hyperparameters are the EGD learning rate $\eta$ and a smoothing factor $\varepsilon$ (0 = no smoothing). For all experiments, we set $\eta=0.01$ and $\varepsilon=1e-3$.

Table 13: **Restricted** hyperparameter values for each data mixing algorithm for experiments where $m\!=\!3$ (corresponding to Table 3 results).

| Data groups | Hyperparameter | Value |
|---|---|---|
| **arXiv/Books/SE** | · proportion of round $\delta$ | 0.288 |
| | · EGD learning rate $\eta$ | 0.2 |
| | · EMA parameter $\gamma$ | None |
| **CommonCrawl/GitHub/Wiki** | · proportion of round $\delta$ | 0.288 |
| | · EGD learning rate $\eta$ | 0.2 |
| | · EMA parameter $\gamma$ | None |

Table 14: **Unrestricted** hyperparameter values for each data mixing algorithm for experiments where $m\!=\!7$ (corresponding to Table 2 results).

| Data groups | Hyperparameter | Value |
|---|---|---|
| **SlimPajama, full** | · proportion of round $\delta$ | 0.07 |
| | · EGD learning rate $\eta$ | 0.2 |
| | · EMA parameter $\gamma$ | 0.1 |

Table 15: **Restricted** hyperparameter values for each data mixing algorithm for experiments where $m\!=\!7$ (corresponding to Table 3 results).

| Data groups | Hyperparameter | Value |
|---|---|---|
| **SlimPajama, full** | · proportion of round $\delta$ | 0.07 |
| | · EGD learning rate $\eta$ | 0.2 |
| | · EMA parameter $\gamma$ | 0.1 |

- **DoGE**: the hyperparameters are the EGD learning rate $\eta$, the smoothing factor $\varepsilon$, and the proportion of the training batch that is allocated for the validation dataset $r$; this is needed to compute the gradient dot-product at each step. We use $\varepsilon\!=\!0$ for all experiments. For $m\!=\!2$, we set $r\!=\!0.25$ and for $m\!=\!3,7$, we set $r\!=\!0.5$. For all experiments besides Github/C4 and SlimPajama, we use $\eta\!=\!0.01$. For Github/C4, we use $\eta\!=\!0.1$ and for SlimPajama we used $\eta\!=\!0.1$ and $\eta\!=\!0.03$ for unrestricted and restricted settings, respectively.

**Weight trajectories.** In Table 16, we provide the mixture proportions for each method (averaged across training steps) for each dataset on one random seed. In Figure 8, we provide all of AIOLI's proportion trajectories throughout training in both the unrestricted and restricted settings on one random seed for the $m\!=\!2$ settings. In Figure 9 and Figure 10, we provide AIOLI's trajectories in the unrestricted and restricted settings on one random seed for Arxiv/Books/StackExchange and CommonCrawl/Github/Wikipedia, respectively. All of our trajectories demonstrate that AIOLI can significantly adjust proportions over time, and that conditioning on different initial proportions can drastically change the behavior of AIOLI.[2]

## F    ADDITIONAL EXPERIMENTS

### F.1    DOWNSTREAM TASKS

We find that lower perplexity is positively correlated with worse performance on downstream tasks. We evaluated all models trained on SlimPajama on ARC-Challenge, ARC-Easy (Clark et al., 2018), BoolQ (Clark et al., 2019), HellaSwag (Zellers et al., 2019), LAMBADA (Paperno et al., 2016), OpenBookQA (Mihaylov et al., 2018), PiQA Bisk et al. (2019), and WinoGrande (Sakaguchi et al., 2020) using the

---

[2]Note that for the restricted setting, AIOLI's trajectory consists of using the base method for a certain amount of steps, and then roughly reverting to the uniform distribution before adjusting the proportions. This is expected behavior, since our initial proportions $p^0$ are uniform in Algorithm 1; this avoids a "biased" proportion update.

Table 16: Average proportions over the entire training trajectory for the unrestricted setting, on one random seed.

| Data groups | Method | Average Proportions |
|---|---|---|
| arXiv/SE | **Grid search** | [0.4, 0.6] |
| | **DML** | [0.404, 0.596] |
| | **Skill-it** | [0.437, 0.563] |
| | **DoReMi** | [0.37, 0.63] |
| | **DoGE** | [0.624, 0.376] |
| | **AIOLI** | [0.507, 0.493] |
| GitHub/C4 | **Grid search** | [0.3, 0.7] |
| | **DML** | [0.46, 0.54] |
| | **Skill-it** | [0.583, 0.417] |
| | **DoReMi** | [0.858, 0.142] |
| | **DoGE** | [0.352, 0.648] |
| | **AIOLI** | [0.505, 0.495] |
| Books/SE | **Grid search** | [0.3, 0.7] |
| | **DML** | [0.381, 0.619] |
| | **Skill-it** | [0.316, 0.684] |
| | **DoReMi** | [0.286, 0.714] |
| | **DoGE** | [0.325, 0.675] |
| | **AIOLI** | [0.456, 0.544] |
| arXiv/Books/SE | **Grid search** | [0.291, 0.306, 0.403] |
| | **DML** | [0.245, 0.277, 0.477] |
| | **Skill-it** | [0.292, 0.238, 0.469] |
| | **DoReMi** | [0.318, 0.180, 0.502] ] |
| | **DoGE** | [0.592, 0.132, 0.276] |
| | **AIOLI** | [0.342, 0.275, 0.383] |
| CC/GitHib/Wiki | **Grid search** | [0.291, 0.306, 0.403] |
| | **DML** | [0.157, 0.472, 0.371] |
| | **Skill-it** | [0.275, 0.3, 0.425] |
| | **DoReMi** | [0.101, 0.714, 0.185] ] |
| | **DoGE** | [0.536, 0.220, 0.244] |
| | **AIOLI** | [0.342, 0.325, 0.333] |
| SlimPajama, full (A/B/C4/CC/G/SE/W) | **Grid search** | [0.202, 0.022, 0.28, 0.038, 0.018, 0.376, 0.064] |
| | **DML** | [0.042, 0, 0, 0.579, 0, 0.249, 0.013] |
| | **Skill-it** | [0.098, 0.111, 0.204, 0.103, 0.138, 0.266, 0.076] |
| | **DoReMi** | [0.08, 0.047, 0.057, 0.11, 0.467, 0.078, 0.157] |
| | **DoGE** | [0.056, 0.162, 0.343, 0.28, 0.038, 0.067, 0.051] |
| | **AIOLI** | [0.142, 0.143, 0.143, 0.144, 0.140, 0.144, 0.143] |

**Language Model Evaluation Harness** (Gao et al., 2024) (Table 17). The correlation between perplexity and the macroaverage of our downstream tasks is 0.529, indicating that lower perplexity is predictive of *worse* downstream performance. In fact, DML obtains the best overall performance, even though it omits three out of seven datasets in SlimPajama (see the average proportions in Table 16).

One potential reason for this disparity is the distribution shift between pre-training data and downstream evaluation data; for example, the DML results suggest that training on Books, C4, and Github is not needed to do well on the above selection of downstream tasks. Many recent works have also noted that perplexity and downstream performance are uncorrelated (Liu et al., 2023; Xia et al., 2023; Tay et al., 2023). Furthermore, Levy et al. (2024) proposes a question answering dataset where the perplexity of the pretrained model is positively correlated with performance, similar to our results. This mismatch between training objective and downstream evaluations also extends to post-training, where better learning of human preferences does not translate to better win-rate against other post-trained models (Chen et al., 2024).

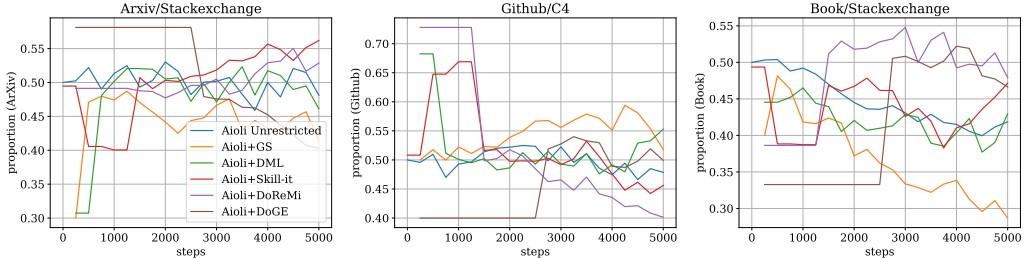

Figure 8: AIOLI's proportions throughout training for both unrestricted and restricted settings on Arxiv/StackExchange, Github/C4, and Book/StackExchange. These trajectories show that AIOLI meaningfully alters the mixture proportions over time.

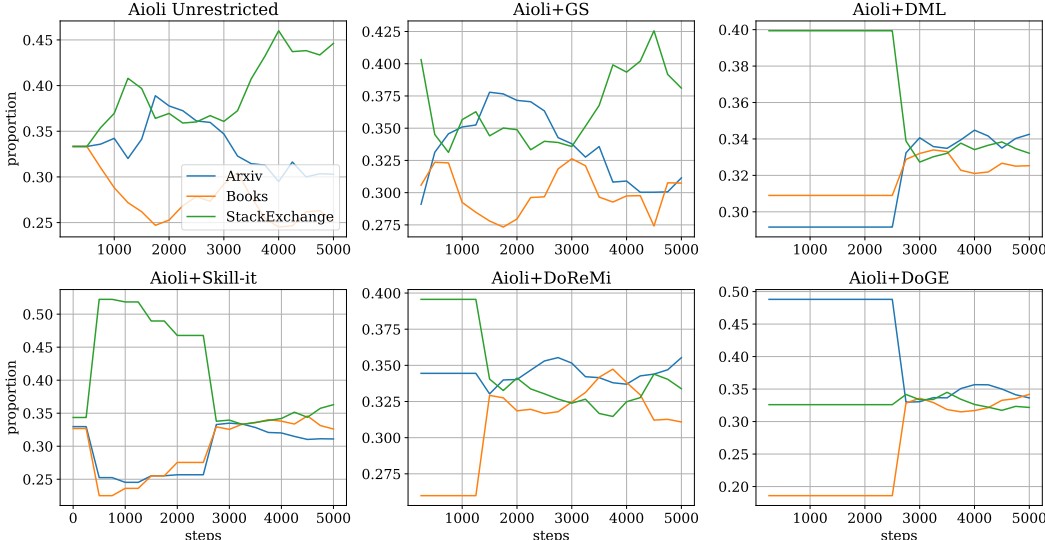

Figure 9: AIOLI's proportions throughout training for both unrestricted and restricted settings on Arxiv/Book/StackExchange.These trajectories show that AIOLI meaningfully alters the mixture proportions over time.

Table 17: Downstream evaluation metrics for various data mixing methods after training on SlimPajama across three random seeds in the unrestricted setting.

| Method | Average | ARC-C | ARC-E | BoolQ | HellaSwag | LAMBADA | OpenBookQA | PiQA | WinoGrande |
|---|---|---|---|---|---|---|---|---|---|
| Stratified | 0.305 | 0.176 | 0.314 | 0.394 | 0.261 | 0.116 | 0.117 | 0.563 | 0.499 |
| AIOLI | 0.311 | 0.172 | 0.315 | 0.447 | **0.264** | 0.114 | 0.111 | 0.559 | **0.504** |
| GS | 0.322 | 0.176 | 0.329 | 0.502 | 0.262 | 0.117 | 0.124 | 0.568 | 0.500 |
| DML | **0.333** | 0.181 | **0.330** | **0.608** | 0.261 | 0.109 | **0.128** | 0.554 | 0.490 |
| Skill-it | 0.316 | **0.182** | 0.322 | 0.462 | 0.261 | 0.124 | 0.122 | 0.559 | 0.492 |
| DoReMi | 0.324 | 0.177 | 0.323 | 0.507 | **0.264** | **0.127** | 0.122 | **0.574** | 0.499 |
| DoGE | 0.314 | 0.173 | 0.313 | 0.471 | 0.262 | 0.116 | 0.115 | 0.557 | **0.504** |

Resolving the disconnect between training objective and downstream evaluations is an area of active research. In the case of data mixing, AIOLI remains the only algorithm in our tests that robustly minimizes average test perplexity–essentially, AIOLI achieves what it sets out to achieve in the LMO framework in (1). Conversely, other data mixing algorithms might be implicitly doing something else with respect to minimizing downstream evaluations. Considering how to incorporate downstream evaluations into data mixing is a fruitful area for future work.

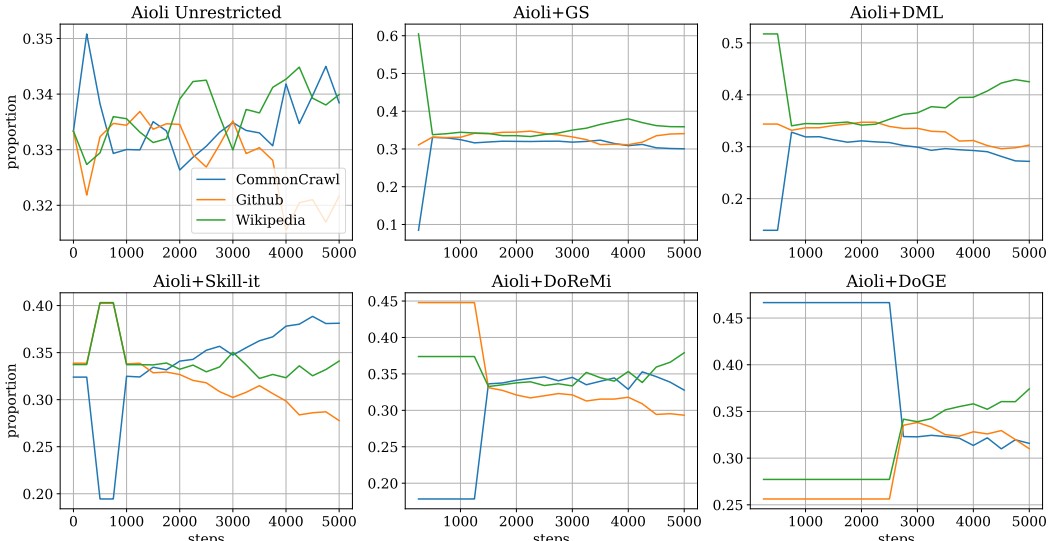

Figure 10: AIOLI's proportions throughout training for both unrestricted and restricted settings on Common-Crawl/Github/Wikipedia.These trajectories show that AIOLI meaningfully alters the mixture proportions over time.

### F.2 ABLATIONS

We ablate AIOLI by studying performance when two key properties of $A^t$ (Appendix C.2.1) are changed: when $T=1$ (i.e., $A^t$ is only learned once at the beginning of training and used throughout), and when $A^t$ is assumed to be diagonal. We evaluate these two ablations in the unrestricted setting presented in Section 6.1 and Table 2:

- AIOLI-STATIC: We set $T=1$ in Algorithm 1. That is, we learn $A^1$ at the beginning of training. We use this $A^1$ to set $p^1$, and use this $p^1$ for the remainder of the training run. This approach tests if $A^t$ needs to be adjusted throughout training.

- AIOLI-DIAGONAL: We assume that each $A^t$ is diagonal in this ablation. In particular, in LEARN-PARAMS we do $A_{ii}^t = \beta_{ii}/p^{t,i}$ rather than $A_i^t = P^{-1}\beta_i$ for each $i \in [m]$ in line 11. This approach tests if it is sufficient to not model cross-group interactions and instead only capture how much group $i$'s performance improves when trained on group $i$ itself.

For both AIOLI-STATIC and AIOLI-DIAGONAL, we use the same set of hyperparameters as AIOLI as described in Appendix E. For AIOLI-STATIC, we additionally sweep over EGD learning rates $\{\eta, 2\eta, 3\eta, 4\eta\}$ where $\eta$ is the EGD learning rate used by AIOLI.

Our results are in Table 18. We find that AIOLI outperforms both ablations in 3 out of 6 settings, and obtains the lowest test perplexity on average over these settings. This suggests that both $T>1$ and modeling off-diagonal entries are important to AIOLI's consistent performance across datasets.

Table 18: Ablations on AIOLI. The table reports the difference in average test perplexity compared to stratified sampling. Negative values (green) = improvement, and bolded = best performing method for given data setting. A=Arxiv, B=Books, GH=GitHub, SE=StackExchange, W=Wikipedia. AIOLI outperforms ablations in 3 out of 6 settings and attains the lowest test perplexity on average.

| Method | A/SE | GH/C4 | B/SE | A/B/SE | CC/GH/W | SlimPajama | Average |
|---|---|---|---|---|---|---|---|
| Stratified | 16.532 | 35.991 | 47.192 | 35.114 | 41.583 | 26.426 | 33.806 |
| AIOLI | **−0.205** | **−0.340** | **−0.439** | −0.226 | −0.196 | −0.240 | **−0.274** |
| AIOLI-STATIC | −0.065 | −0.333 | −0.226 | −0.117 | 0.092 | **−0.330** | −0.140 |
| AIOLI-DIAGONAL | −0.182 | −0.178 | −0.354 | **−0.246** | **−0.215** | −0.202 | −0.230 |

## F.3 HYPERPARAMETER SENSITIVITY

We study how robust AIOLI is to changes in its hyperparameters. From the experimental details in Appendix E, the main hyperparameters that we modify are $\eta$ (EGD step size), $\delta$ (proportion of round allocated for learning $A^t$), and $\gamma$ (the EMA parameter). In Tables 19, 20, and 21, we report results on AIOLI in the unrestricted setting for Arxiv/StackExchange and Arxiv/Books/StackExchange. We sweep $\eta \in \{0.1, 0.2, 0.3, 0.5\}$, $\delta/m \in \{0.064, 0.096, 0.128\}$, and $\gamma \in \{\text{None}, 0.1, 0.5\}$. We find that AIOLI still yields lower test perplexity than stratified sampling across all $\eta, \delta$, and $\gamma$ we evaluated.

Table 19: The difference in average test perplexity of AIOLI with varying $\eta$ step size hyperparameter compared to stratified sampling. Bolded result is the original number reported in Table 2.

| Method | A/B | A/B/SE |
|---|---|---|
| Stratified | 16.532 | 35.114 |
| AIOLI ($\eta=0.1$) | $-0.110$ | $-0.212$ |
| AIOLI ($\eta=0.2$) | $\mathbf{-0.205}$ | $-0.221$ |
| AIOLI ($\eta=0.3$) | $-0.155$ | $-0.186$ |
| AIOLI ($\eta=0.5$) | $-0.166$ | $\mathbf{-0.226}$ |

Table 20: The difference in average test perplexity of AIOLI with varying $\delta/m$, the fraction of each round for learning $A^t$, compared to stratified sampling. Bolded result is the original number reported in Table 2.

| Method | A/B | A/B/SE |
|---|---|---|
| Stratified | 16.532 | 35.114 |
| AIOLI ($\delta/m=0.064$) | $\mathbf{-0.205}$ | $-0.152$ |
| AIOLI ($\delta/m=0.096$) | $-0.283$ | $\mathbf{-0.226}$ |
| AIOLI ($\delta/m=0.128$) | $-0.003$ | $-0.296$ |

Table 21: The difference in average test perplexity of AIOLI with varying $\gamma$, the hyperparameter for computing $p^t$ with an exponential moving average, compared to stratified sampling. Bolded result is the original number reported in Table 2.

| Method | A/B | A/B/SE |
|---|---|---|
| Stratified | 16.532 | 35.114 |
| AIOLI ($\gamma=\text{None}$) | $-0.11$ | $\mathbf{-0.226}$ |
| AIOLI ($\gamma=0.1$) | $\mathbf{-0.205}$ | $-0.185$ |
| AIOLI ($\gamma=0.5$) | $-0.141$ | $-0.213$ |

## F.4 RESULTS ON LARGER MODELS

We examine if our findings—both in terms of the mixing law and in terms of AIOLI's performance—hold on larger models. We train 1.4B-parameter models. We use a learning rate of `3e-4` and keep all other training details the same. We use a subsample of our data settings, focusing on when we mix Arxiv/StackExchange ($m=2$) and Arxiv/Book/StackExchange ($m=3$).

First, we measure if the log-linear static and linear-dynamic mixing laws are well-specified for 1.4B models. We use the same fitting procedure as described in Section 4.1 and Appendix C.1. Figure 11 describes the fit of the static and dynamic mixing laws on Arxiv/StackExchange. The full results are in Table 22, which show that the average $R^2$ for the static and dynamic mixing laws for the 1.4B model are 0.989 and 0.929, respectively. This accuracy of the mixing law parameterization on the 1.4B model is a prerequisite for AIOLI's performance, which we evaluate next.

Second, we evaluate AIOLI in the unrestricted setting on the 1.4B models. We compare AIOLI to stratified sampling and DoGE. Our results on three random seeds are in Table 23. Similar to our results on the 160M models, we find that AIOLI outperforms stratified sampling in both data settings. Moreover, from Table 2, we see that DoGE originally performed worse than stratified sampling at the 160M scale. Our results here confirm that even at the 1.4B model scale, DoGE continues to underperform stratified sampling. Altogether, we see that AIOLI consistently outperforms stratified sampling while existing methods do not—at both the 160M and 1.4B scale.

Table 22: Comparison of log-linear static and linear dynamic mixing law parameterizations when training a 1.4B model.

| Parameterization | Arxiv/SE | | Arxiv/Books/SE | |
|---|---|---|---|---|
| | MSE | $R^2$ | MSE | $R^2$ |
| Log-linear static | 2e-4 | 0.995 | 1e-3 | 0.984 |
| Linear dynamic | 7e-5 | 0.916 | 2e-4 | 0.943 |

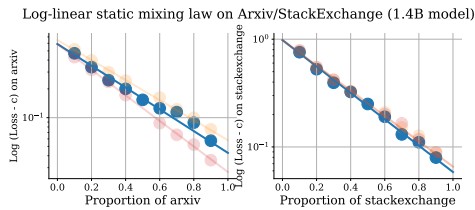
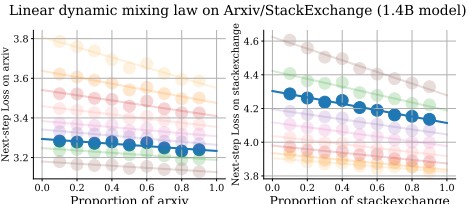

Figure 11: Left: log-linear static mixing law fit on Arxiv/Stackexchange on 1.4B parameter model, in which each color represents a different random seed. Right: linear dynamic mixing law fit on Arxiv/Stackexchange on 1.4B parameter model on 1 random seed. Each color is a different initial mixture $p^0 \in \mathcal{P}$ trained for 2000 steps, and the fitting sweeps are done over 100 additional steps.

Table 23: Difference in average test perplexity compared to stratified sampling in the unrestricted setting for 1.4B models. For AIOLI, we use $\eta=0.5, \delta/m=0.096, \gamma=0.1$ for A/SE and $\eta=0.1, \delta/m=0.096, \gamma=0.5$ for A/B/SE.

| Method | A/SE | A/B/SE |
|---|---|---|
| Stratified | 15.799 | 34.733 |
| DoGE | 0.551 | 0.922 |
| AIOLI | −0.276 | −0.403 |

## F.5 OUT-OF-DOMAIN SETTING

We consider the *out-of-domain* setting, in which the training data groups are disjoint from the groups that the model will be evaluated on. This is a practical scenario where we have access to a separate validation dataset that we wish our model to perform well on (Fan et al., 2024; Chen et al., 2023; Xia et al., 2024; Xie et al., 2023b; Engstrom et al., 2024). We will demonstrate how 1) the LMO framework can be adjusted to capture this setting, recovering the out-of-domain versions of Skill-It and DoGE proposed in their respective papers; 2) the linear mixing laws are still well-specified in this setting; and 3) AIOLI adjusted for this setting can still more consistently outperform out-of-domain baselines.

**LMO framework for OOD setting.** Concretely, we suppose we have $m$ training data groups such that $D_{\text{train}}$ is still $\{D_{\text{train}}^1,...,D_{\text{train}}^m\}$, and we have one separate out-of-domain data group that we do not train on; we have IID validation and test datasets for this out-of-domain data group. Let $L_{\text{val, OOD}}$ be the validation and test loss on the out-of-domain data group, respectively. Then, the LMO framework can be slightly modified:

$$\text{minimize}_{\boldsymbol{p} \in \triangle^{T \times m}} L_{\text{val, OOD}}^{T+1}(\boldsymbol{p}) \tag{52}$$

$$\text{s.t. } L_{\text{val, OOD}}^{t+1}(\boldsymbol{p}) = c^t + b^t \sigma\left(\sum_{j=1}^m -A_{\text{OOD},j}^t p_j^t\right) \forall t \in [T], \tag{53}$$

where $A_{\text{OOD},j}^t \in \mathbb{R}^m$ is now a vector representing how much each training group influences the validation group. There are two changes to the optimization problem: first, the objective is now to minimize the out-of-domain validation loss; second, the mixing law captures the relationship between the validation loss and the mixture proportions over the training data groups. Note that the DML method can still be applied in the OOD setting by directly minimizing $c + b \exp(\sum_{j=1}^m -A_{\text{OOD},j} p_j)$. More importantly, applying

Lemma 1 to this optimization problem, we get the update rule $p_j^{t+1} \sim p_j^t \exp(\eta A_{\text{OOD},j}^t) \, \forall j \in [m]$. This expression recovers the Skill-It and DoGE OOD update rules, and can be incorporated into AIOLI as demonstrated in Algorithms 3 and 4. These algorithms are identical to AIOLI (Alg 1) and LEARNPARAMS (Alg 2), with the exception of lines 6 and lines 3, 8, and 11 respectively, which reflect that $A_{\text{OOD}}^t$ is now a vector rather than an $m \times m$ matrix.

---

**Algorithm 3** AIOLI-OOD

---

1: **Input:** data $D_{\text{train}}$, $D_{\text{val}}$, model $f^1$. Initial steps $S_{\text{init}}$, initial proportions $\boldsymbol{p}^{\text{init}} \in \triangle^m$. $T$ rounds over $S - S_{\text{init}}$ remaining steps, $\delta$ fraction per round for learning parameters, learning rate $\eta$, one-hot smoothing factor $\varepsilon$.
2: If $S_{\text{init}} \neq 0$, train $f^1$ on $\boldsymbol{p}^{\text{init}}$ for $S_{\text{init}}$ steps.
3: Set $p^0 = \text{Unif}(m)$.
4: **for** $t = 1, ..., T$ **do**
5:     Set $A_{\text{OOD}}^t, f^{t+\delta} \leftarrow \text{LEARNPARAMS-OOD}(D_{\text{train}}, D_{\text{val}}, \delta, f^t, \varepsilon)$ (Alg. 4), and normalize $A^t$ to get $\bar{A}^t$.
6:     $p_j^t \propto p_j^{t-1} \exp(\eta \bar{A}_{\text{OOD},j}^t)$ for all $j \in [m]$.
7:     Train model $f^{t+\delta}$ with $\frac{S}{T}(1-\delta)$ steps from mixture $p^t$ over $D_{\text{train}}$. Obtain updated $f^{t+1}$.
8: **end for**

---

**Algorithm 4** LEARNPARAMS-OOD

---

1: **Input:** $D_{\text{train}}$, $D_{\text{val}}$, $\delta$, model $f^t$, number of sweeps $k$, one-hot smoothing factor $\varepsilon$.
2: Split the fraction of a training round $\delta$ into $K$ time segments, where $K = mk$.
3: Set $\beta = \vec{0} \in \mathbb{R}^m$.
4: Define $p^{t,i} = (1-\varepsilon)\mathbf{1}_i + \varepsilon \text{Unif}(m)$ for $i \in [m]$, and define $P = [p^{t,1}, ..., p^{t,m}] \in \triangle^{m \times m}$.
5: Randomly shuffle $k$ instances of each $i \in [m]$ to create an order $\mathcal{I} \in [m]^K$.
6: **for** $\tau = 1, ..., K$ **do**
7:     Let $j = \mathcal{I}_\tau$. Train model on mixture $p^{t,j}$ of $D_{\text{train}}$ for one time segment, obtain $f^{t+\tau\delta/K}$.
8:     Update $\beta_j \leftarrow \beta_j + L_{\text{val,OOD}}(f^{t+(\tau-1)\delta/K}) - L_{\text{val,OOD}}(f^{t+\tau\delta/K})$ with loss difference on OOD validation dataset.
9: **end for**
10: Update $\beta \leftarrow \frac{\beta}{k}$.
11: Set $A_{\text{OOD}}^t = P^{-1}\beta$.
12: **Return** $A_{\text{OOD}}^t \in \mathbb{R}^m, f^{t+\delta}$

---

**Mixing law parameterization results.** We study a setting where our training data groups are Arxiv, Book, and Github from SlimPajama and our validation data group is StackExchange. Using the same setup as other $m = 3$ settings in Section 4.2 (160M model, 5K steps, sweep over 9 runs), we measure the MSE and $R^2$ of the log-linear static mixing law, $L_{\text{val, OOD}}(\boldsymbol{p}) = c + b \exp\left(\sum_{j=1}^m -A_{\text{OOD},j}p_j\right)$, and of the linear dynamic mixng law, $L_{\text{val, OOD}}^{t+1}(\boldsymbol{p}) = c^t + b^t \sum_{j=1}^m -A_{\text{OOD},j}^t p_j^t$. The MSE and $R^2$ for the log-linear static mixing law are $1.5 \times 10^{-3}$ and $0.964$, respectively. The MSE and $R^2$ for the linear dynamic mixing law are $1.1 \times 10^{-4}$ and $0.796$. The linear dynamic mixing law fits the true loss-proportion relationship less accurately than the log-linear static law. Nevertheless, both MSEs are low, and the $R^2$ still suggests that at least $79\%$ of the variability in validation loss can be explained by the mixing law.

**AIOLI results.** We evaluate stratified sampling, and OOD versions of AIOLI, Skill-It, DoGE, and DML in the unrestricted setting on 3 random seeds. We train on Arxiv, Books, and Github and evaluate on StackExchange. Our results are in Table 24.

We find that all methods, including AIOLI, attain lower test perplexity than the stratified sampling baseline, which both the Skill-It and DoGE papers use as a comparison point for the OOD setting. AIOLI is the only method that achieves this improvement without requiring additional training runs. This improvement over stratified sampling across OOD methods is expected, since stratified sampling can include irrelevant data

Table 24: Out-of-domain data evaluation, in which we mix training data from Arxiv, Books, and Github and evalute on StackExchange data. The table reports the difference in average test perplexity compared to stratified sampling on the training data groups. For AIOLI, we use $\eta = 0.8, \delta/m = 0.096, \gamma =$ None.

| Method | Arxiv/Book/Github $\rightarrow$ StackExchange | # extra runs |
|---|---|---|
| Stratified | 39.644 | 0 |
| GS | $-7.244$ | 10 |
| DML | $-6.316$ | 10 |
| Skill-It (OOD) | $-5.786$ | 3 |
| DoGE (OOD) | $-7.626$ | 1 |
| AIOLI (OOD) | $-4.028$ | 0 |

due to the distribution shift between training and evaluation. On the other hand, stratified sampling is a strong baseline in the in-distribution scenarios studied in the rest of this work.

## G    ADDITIONAL DISCUSSION

**Hyperparameter Optimization and Truncation Bias.** Many data mixing methods utilize extra training runs to learn the static mixture proportions before the final training run. This allows us to view data mixing as a hyperparameter optimization problem in $p$. Ye et al. (2024) and Liu et al. (2024) mitigate the inefficiency of grid search in higher dimension by combining it with data mixing laws to impose additional structure. By fitting the loss surface, these methods obtain better sample complexity and performance than grid search alone. However, both grid search and these offline methods can have poor performance when $p$ is searched for or fitted on shorter runs, as in the restricted setting.

To understand these results, we look to the hyperparameter optimization literature. Many popular hyperparameter optimization methods carefully control truncation, and some runs are allowed to continue longer than others (Li et al., 2018; Swersky et al., 2014; Domhan et al., 2015). Thus, generic hyperparameter optimization methods may eventually prove effective for tuning data mixes, but for future work we recommend going beyond a fixed compute budget for all runs in a hyperparameter sweep, especially in compute-constrained settings.

## H    WHY THE METHOD IS CALLED AIOLI

An aioli is an emulsion, where individual components remain chemically separate from each other, despite being combined into one mixture. Similarly, our $A^t$ matrix is formed from separate test runs (the $p^{t,1}, \dots, p^{t,m}$ in Section 5), despite being combined into one update for $p^t$.

