# OpenReview forum: "Aioli: A Unified Optimization Framework for Language Model Data Mixing"
_ICLR.cc/2025/Conference — ICLR 2025 Poster_

### Official Review · Reviewer_9HhZ · 2024-10-29

**Soundness:** 3
**Presentation:** 3
**Contribution:** 3
**Rating:** 6
**Confidence:** 4

**Summary:**

This paper unifies all existing data mixture optimization methods within one Linear Mixing Optimization (LMO) framework, and proposes AIOLI as a new online data mixing method. AIOLI iteratively train models by upweighting each training data domains and leverages the loss difference to learn the optimal data mixing coefficients.
Empirically, AIOLI is able to consistently outperform or be comparable to the stratified sampling baseline in the unrestricted setting, without extra proxy runs. Additionally, AIOLI can improve on the existing online data mixing methods by a significant margin.

**Strengths:**

1. The paper introduce a novel online data mixing algorithm AIOLI leveraging the EGD optimization framework, without introducing extra runs as previous methods.
2. The paper also conclude all existing data mixing methodologies within an unified optimization framework, and conducted a comprehensive evaluation with the target of minimizing the average domain losses.
3. The proposed method AIOLI demonstrates remarkable robustness comparing to all the other baseline methods, which not requires extra runs with proxy models.

**Weaknesses:**

1. **The proposed algorithm is very similar to the Skill-it method [1]**, which weakens the algorithmetic novelty.
- The `LearnParams` algorithm (alg. 2) in the AIOLI paper is very similar to the `LearnGraph` algorithm (Alg. 3 [1]) from the Skill-it paper, where the adjacency matrix A has been updated according to the validation loss difference.  Each "training domain" in AIOLI can be considered as each "skill" node in the graph, which corresponds to the *Continual pre-training* setting in [1], Table1.
- The `AIOLI` algorithm (alg.1) is also similar to the `SKILL-IT Online Data Selection Algorithm` (alg. 1, [1]), which applies EGD on the data mixture weights. The author considers this as an unified online optimization framework for data mixture optimization problem.

2. The paper only targets at minimizing the average losses but not consider the other evaluation cases (e.g. out-of-domain genearlization etc). How would AIOLI perform comparing to the other baselines when the validation distribution is unknown (e.g. the OoD setting in DOGE paper [2] and the Skill-it paper [1])?

3. The finalized domain weights from AIOLI are very close to uniform (Table16), which may explain the consistently comparable-to-baseline results in Fig. 3.  Since the uniform sampling across difference domains would be the first-choice for practitioners for minimizing the average domain loss, how much the AIOLI framework could help with LLM pretraining above uniform-sampling baseline in practice?

[1] Skill-it! A Data-Driven Skills Framework for Understanding and Training Language Models

[2] DoGE: Domain Reweighting with Generalization Estimation

**Questions:**

1. When the validation distribution is unknown or different from the training domains (e.g. train on the other 6 domains but eval on Books/Wiki/Github etc.)?

2. Does the proposed method robust about the hyperparameter k, $\delta$ (in Algorithm2)? To understand their impact, could you provide ablation studies or sensitivity analyses for these specific hyperparameters?

3. The finalized domain weights from AIOLI are very close to uniform (Table16), how much the AIOLI framework could help with LLM pretraining above uniform-sampling baseline in practice?

4. Could you show the data mixture weights trajectory from AIOLI? Does it learn from the curriculum like in DOREMI [1] and DOGE [2]  or it is always close to uniform?

[1] DoReMi: Optimizing Data Mixtures Speeds Up Language Model Pretraining

[2] DoGE: Domain Reweighting with Generalization Estimation

---

### Official Review · Reviewer_HM3q · 2024-11-02

**Soundness:** 4
**Presentation:** 4
**Contribution:** 3
**Rating:** 6
**Confidence:** 4

**Summary:**

This paper unifies existing methods for data mixing, offline and online, as different parameter estimations of mixing laws. From this perspective, the authors find existing methods to fall short because of inaccurate parameter estimations. With this insight, the author derives a new online data mixing method AIOLI, which updates data proportion according to dynamically fitted data mixing laws along the training process. Experiment results show AIOLI generally works well on 6 out of 6 datasets while other methods underperform some of them.

**Strengths:**

1. The paper is well-organized and easy to follow, with extensive experimental details.
2. This paper provides a unified framework for data mixing problems that explain the different of different methods, which is a significant conceptual contribution.
3. The experiment results confirm the method generally works well, shows improvement on 6 out of 6 datasets,

**Weaknesses:**

1. Although AIOLI improves stratified sampling on all evaluated data, it can underperform baseline methods in some datasets (e.g., A/B/SE in Tab. 2).
2. The authors propose the LearnParams Algorithm to estimate A* without sweeping, but its accuracy is unclear. It is also unclear whether the simulating process affects model performance.
3. AIOLI involves a number of hyperparameters while a description of how to decide these hyperparameters is insufficient. This may hinder the application AIOLI.

**Questions:**

AIOLI only performs one update step on each round (Line 5 in Algorithm 1). Why not iterate for a few steps to approach the optimal mixture proportion given by the estimated mixing laws?

---

### Official Review · Reviewer_aN2q · 2024-11-04

**Soundness:** 2
**Presentation:** 3
**Contribution:** 3
**Rating:** 5
**Confidence:** 4

**Summary:**

The research content of this paper is about how to optimize the data mixing of pre-training corpus. This paper proposes a framework for mixture proportions--LMO, which can incorporate existing methods (this paper lists one offline and three online methods) into this framework. The basic assumption (loss-proportion relationship) of this framework is verified through experiments, and the shortcomings of parameter estimation in existing methods (mainly three offline methods) are pointed out (disparity between estimated parameters and Optimal). This paper improved the parameter estimation. Experimental results have shown that the improved method proposed in this paper can enhance the performance on its test set across two situations.

**Strengths:**

This article is well written and establishes a mathematical expression framework that unifies the data mixing methods mentioned in this paper. It reveals that the different data mixing methods mentioned in this paper are formally unified, with differences mainly concentrated in parameter estimation methods. At the same time, a targeted method for parameter estimation in the framework was proposed, which demonstrated effectiveness in practical experiments.

**Weaknesses:**

I am willing to acknowledge the contribution of the framework proposed in this paper, but the proposal of the framework focuses more on "induction" based on existing methods, which leads to a lack of persuasiveness in the innovation and practicality of the framework. Specifically, 1) the basic assumption of the framework is that the loss-proportion relationship is "linear" (or log linear). Although there has been experimental verification, the experiments are very empirical and the experimental setup is relatively simple, involving up to 7 data domains (real pre-training scenarios often contain domains of over 10000 levels). 2) The validation was only conducted at 160M (section 4.1), which is not convincing at all (many abilities cannot emerge on models of this magnitude), especially considering the extensive research on Scaling laws, small models can be used to predict larger models; 3) This paper only needs to consider the impact between up to 2 data domains, and characterizing the impact between more data is limited. However, this is more of a concern in real pre-training scenarios; 4) The parameter improvement method proposed in this paper is also biased towards empirical methods (line 378-line 389), and lacks persuasiveness in terms of generality and effectiveness on a wider scale。

**Questions:**

1.In Figure 2, the author used all the points for fitting. Did the author use the fitted mixing data law to predict a certain proportion setting?
2. May I ask if the author has conducted experiments on larger models or have studied the argument that the conclusions drawn in small models also hold true in larger models?
3. Would the author mind explaining in layman's terms the basic principles of the effectiveness of Algorithm 2 or the basic principles of the improvement brought by this method (line379-line387) (or whether this method is a universal approach);
4. Besides PPL, may I ask if the author has tested the basic performance of the model downstream under different data mixing strategies?

---

### Official Review · Reviewer_zQiM · 2024-11-04

**Soundness:** 3
**Presentation:** 3
**Contribution:** 4
**Rating:** 8
**Confidence:** 4

**Summary:**

This paper targets the challenge of optimizing data mixture groups in training large language models. The authors claims that no existing method consistently outperforms a simple baseline regarding average test perplexity. They propose a unified optimization framework that encompasses several recent methods, demonstrating that while these methods can represent the true loss-proportion relationship, they often fail to set accurate parameters, leading to sub-optimal performance. Leveraging this framework, the authors introduce a new method, AIOLI, which consistently surpasses baselines in both general and practical settings.

**Strengths:**

1 The unified framework integrates several recent methods about data mixture on large language model. This part is well-written and easy to follow.
2 The problem it tries to tackle is of great importance. Related work propose different methods, and this paper try to unify them as a framework, which could advance future research in this topic.
3 The experiments provided are all necessary to support the claims.

**Weaknesses:**

1 Formulas and letters and extensive details make some contents not easy to follow. For instance, in section 5 (Estimating $A^{t\star}$), additional clarification on the core concepts prior to presenting numerous notations would enhance understanding.
2 Experimental settings are insufficient. All experiments utilize a 160M model with a maximum of 50K steps. In Section 4.1, when m=7, 10 different proportions are not enough to find the optimal proportion.

**Questions:**

1 How do you guarantee the "LearnParams" will yield accurate estimate given its parameters and potential for error?
2 The paper employs "gains in every data group" as a metric to demonstrate Aioli's effectiveness. However, in practical scenarios, a model need not outperform its counterparts in every data group. How would you address situations where different weights are assigned to various data groups?

---

### Meta-Review · Area_Chair_Fc92 · 2024-12-21

**Metareview:**

The paper introduces a framework that unifies various existing data mixing methods, providing a theoretical lens to analyze their strengths and weaknesses. The algorithm demonstrates consistent improvement over stratified sampling and robustness across different datasets. Data mixing is a crucial aspect of training large language models, making the findings directly applicable to real-world scenarios. One reviewer pointed out similarities between Aioli and the Skill-It method, particularly in the algorithms for updating the adjacency matrix and applying EGD. The authors clarified the differences in motivation and parameter estimation, but some concerns remained regarding the degree of novelty. The authors provided comprehensive experiments and addressed reviewer concerns effectively in the rebuttal. The paper's strengths, particularly the LMO framework and the consistent performance of Aioli, likely outweigh the weaknesses. The concerns regarding practicality and hyperparameter tuning are important but could potentially be addressed in future work or through further clarification. The paper is marginally above the acceptance threshold based on the provided information.

**Additional Comments On Reviewer Discussion:**

Reviewer aN2q, who self-identified as having extensive pre-training experience, expressed doubts about the practicality of Aioli and data mixing methods in general, based on their own experience. This suggests a potential gap between the theoretical improvements and real-world applicability. The authors tried to clarify the practicality of Aioli given it does not require any extra training runs, unlike other methods. Reviewer aN2q mentioned that they often manually adjust sampling ratios based on downstream capabilities. A comparison to such manual strategies might provide additional insights.

---

### Decision · Program_Chairs · 2025-01-22

Accept (Poster)